# Sparse-MoE-SAM: A Lightweight Framework Integrating MoE and SAM with a Sparse Attention Mechanism for Plant Disease Segmentation in Resource-Constrained Environments

**DOI:** 10.3390/plants14172634

**Published:** 2025-08-24

**Authors:** Benhan Zhao, Xilin Kang, Hao Zhou, Ziyang Shi, Lin Li, Guoxiong Zhou, Fangying Wan, Jiangzhang Zhu, Yongming Yan, Leheng Li, Yulong Wu

**Affiliations:** 1School of Electronic Information and Physics, Central South University of Forestry and Technology, Changsha 410004, China; 20223337@csuft.edu.cn (B.Z.); 20231100403@csuft.edu.cn (H.Z.); 20231200582@csuft.edu.cn (Z.S.); t19940562@csuft.edu.cn (F.W.); t20070605@csuft.edu.cn (J.Z.); t20080581@csuft.edu.cn (Y.Y.); 2School of Computer, Jiangsu University of Science and Technology, Zhenjiang 212100, China; 232241807217@stu.just.edu.cn; 3School of Forestry, Central South University of Forestry and Technology, Changsha 410004, China; 20220147@csuft.edu.cn; 4Bangor College, Central South University of Forestry and Technology, Changsha 410004, China; t20216055@csuft.edu.cn

**Keywords:** plant disease segmentation, sparse attention, mixture of experts, SAM (Segment Anything Model)

## Abstract

Plant disease segmentation has achieved significant progress with the help of artificial intelligence. However, deploying high-accuracy segmentation models in resource-limited settings faces three key challenges, as follows: (A) Traditional dense attention mechanisms incur quadratic computational complexity growth (O($n^{2}d$)), rendering them ill-suited for low-power hardware. (B) Naturally sparse spatial distributions and large-scale variations in the lesions on leaves necessitate models that concurrently capture long-range dependencies and local details. (C) Complex backgrounds and variable lighting in field images often induce segmentation errors. To address these challenges, we propose Sparse-MoE-SAM, an efficient framework based on an enhanced Segment Anything Model (SAM). This deep learning framework integrates sparse attention mechanisms with a two-stage mixture of experts (MoE) decoder. The sparse attention dynamically activates key channels aligned with lesion sparsity patterns, reducing self-attention complexity while preserving long-range context. Stage 1 of the MoE decoder performs coarse-grained boundary localization; Stage 2 achieves fine-grained segmentation by leveraging specialized experts within the MoE, significantly enhancing edge discrimination accuracy. The expert repository—comprising standard convolutions, dilated convolutions, and depthwise separable convolutions—dynamically routes features through optimized processing paths based on input texture and lesion morphology. This enables robust segmentation across diverse leaf textures and plant developmental stages. Further, we design a sparse attention-enhanced Atrous Spatial Pyramid Pooling (ASPP) module to capture multi-scale contexts for both extensive lesions and small spots. Evaluations on three heterogeneous datasets (PlantVillage Extended, CVPPP, and our self-collected field images) show that Sparse-MoE-SAM achieves a mean Intersection-over-Union (mIoU) of 94.2%—surpassing standard SAM by 2.5 percentage points—while reducing computational costs by 23.7% compared to the original SAM baseline. The model also demonstrates balanced performance across disease classes and enhanced hardware compatibility. Our work validates that integrating sparse attention with MoE mechanisms sustains accuracy while drastically lowering computational demands, enabling the scalable deployment of plant disease segmentation models on mobile and edge devices.

## 1. Introduction

Plant diseases present a persistent threat to global agricultural productivity, with profound implications for food security, economic stability, and environmental sustainability [1]. Conventional disease identification predominantly relies on manual inspection and expert diagnosis, which are labor-intensive, time-consuming, and difficult to scale, while automated computer vision methods for disease detection and segmentation have gained traction with increasing access to high-resolution imagery and computational resources [2,3], achieving precise and efficient diseased-region segmentation remains challenging. This is attributable to symptom heterogeneity, imaging condition variability, and the complex structural features of plant leaves.

Established deep learning models—including Convolutional Neural Networks (CNNs) and specialized architectures like U-Net and DeepLab—have demonstrated strong performance in generic image segmentation [4,5,6]. However, their utility in practical plant disease segmentation is limited by three factors [7]. First, computational bottlenecks arise from intensive feature processing pipelines, restricting real-time deployment on edge devices in agricultural environments [8]. Second, insufficient domain adaptation in model architectures impedes generalization across diverse plant species, disease types, and environmental conditions. Third, overfitting to training data distributions reduces robustness against natural variability in field-collected imagery.

Recent Transformer-based advances leverage global self-attention to enhance modeling capacity [9,10,11]. Foundational models like Meta’s Segment Anything Model (SAM) exhibit exceptional zero-shot generalization across segmentation tasks. Contemporary research has explored various efficiency improvements for plant disease segmentation. For instance, recent work by Upadhyay et al. (2025) reviewed advancements in deep learning techniques, emphasizing the use of convolutional neural networks (CNNs) and transformers for disease detection [12]. Zhang et al. (2025) proposed Reformer with re-parameterized kernels for grape disease segmentation, achieving competitive accuracy with reduced parameters [13]. Hu et al. (2024) introduced the LVF framework combining language and vision features for tomato disease segmentation, demonstrating multimodal fusion benefits [11]. However, these approaches either focus on specific crops or lack comprehensive efficiency optimization for edge deployment.

Nonetheless, agricultural deployment faces critical limitations [14]. SAM’s dense attention layers, although effective, incur quadratic computational complexity (O(n^2^)) relative to input size, making them unsuitable for deployment on resource-constrained agricultural hardware. Recent efficiency-oriented solutions include sparse attention patterns [15], pruning strategies [16], and knowledge distillation [17], yet none specifically address the unique challenges of agricultural deployment: extreme computational constraints, diverse environmental conditions, and the inherent sparsity of disease symptoms.

Comparison with Non-SAM Sparse Attention Architectures: Beyond the SAM series, several sparse attention mechanisms have been developed for computer vision tasks. Linformer reduces attention complexity to O(nk) through low-rank matrix approximation but lacks content-aware selection crucial for disease symptom localization. Performer employs random feature maps for linear attention complexity but struggles with precise boundary delineation required in medical/agricultural applications. Swin Transformer utilizes windowed attention for computational efficiency but its fixed spatial partitioning cannot adapt to the irregular, multi-scale nature of plant disease patterns. PVT (Pyramid Vision Transformer) incorporates spatial-reduction attention but maintains dense processing within reduced regions, limiting efficiency gains. Our Gumbel-TopK approach differs fundamentally by: (1) content-adaptive selection vs. fixed patterns, (2) differentiable routing enabling end-to-end training vs. discrete approximations, and (3) biologically motivated sparsity patterns specifically designed for agricultural pathology vs. general computer vision tasks. Comparative experiments show our method outperforms Linformer by 3.7% IoU, Performer by 2.9% IoU, and Swin-S by 1.8% IoU while achieving comparable or superior computational efficiency.

Plant diseases typically manifest as localized patterns (e.g., small lesions, chlorotic patches, irregular discolorations) exhibiting inherent spatial sparsity and multi-scale properties. This biological insight motivates architectures that prioritize computation for salient regions while suppressing irrelevant background features [15,16]. An adaptive mechanism enabling dynamic allocation of computational resources to discriminative feature domains based on spatial saliency is thus essential.

Efficient attention mechanisms and expert specialization represent critical design paradigms for addressing these challenges. Sparse attention architectures have demonstrated effectiveness in reducing computational complexity while preserving semantic relationships, particularly in scenarios with inherent spatial sparsity. Expert specialization through MoE frameworks enables adaptive processing tailored to distinct input characteristics. In plant pathology specifically, recent works have explored efficiency improvements: Guo et al. (2024) investigated dual U-shaped networks with coordinate attention for pest segmentation [18], while Yang et al. (2024) applied multi-scale attention for lesion segmentation [7]. However, these approaches primarily focus on conventional attention mechanisms without addressing the fundamental quadratic complexity issue. Sparse attention applications in agricultural computer vision remain largely unexplored, despite the natural sparsity of disease symptoms presenting ideal opportunities for computational optimization. Similarly, while MoE architectures have shown promise in natural language processing and general computer vision, their application to plant disease segmentation—where symptoms exhibit diverse morphological patterns requiring specialized processing—represents a significant research gap. The unique challenges of agricultural deployment, including extreme computational constraints, diverse environmental conditions, and the inherent sparsity of disease manifestations, necessitate novel architectural solutions that jointly optimize accuracy and efficiency.

Effective deployment in agricultural settings requires an optimal trade-off between segmentation accuracy, computational efficiency, and robustness to imaging variations. This work establishes a framework that advances state-of-the-art benchmark performance while ensuring minimal accuracy loss when transferred to field conditions [19].

This paper introduces Sparse-MoE-SAM, a novel architectural paradigm that represents the first comprehensive integration of bio-inspired sparse attention with hierarchical mixture-of-experts specifically designed for agricultural computer vision. Unlike existing approaches that address computational efficiency or segmentation accuracy in isolation, our framework simultaneously achieves both objectives through three synergistic innovations:(a)To mitigate computational overhead, we introduce a sparse attention mechanism integrated with the MoE framework, further evaluated in the experimental section, we introduce a spatially constrained sparse attention module. This module employs dynamic top-k selection, retaining only the top k% of attention weights (where k ≪ n), thereby reducing self-attention complexity to O(nk). This design reflects the natural sparsity patterns observed in plant lesions, such as localized necrotic tissue, achieving strong alignment with pathologist annotations (Pearson r > 0.82). The mechanism preserves over 90% of critical feature connections while achieving a 23.7% reduction in computational requirements, enabling efficient processing of high-resolution field imagery.(b)To effectively process sparsely distributed, multi-scale lesions that require integration of both local and global contextual information, we develop a dual-stage MoE framework. Coarse-grained segmentation and fine-grained refinement are performed by specialized expert networks tailored to each processing stage. A gating network dynamically routes features based on input complexity, enabling this cascaded architecture to achieve superior boundary discrimination compared to uniform processing approaches, particularly under cluttered background conditions.(c)To mitigate segmentation errors arising from complex backgrounds and varying illumination conditions, we integrate sparse attention mechanisms into an enhanced ASPP module. This design captures multi-scale contextual information across diverse lesion sizes, from minute lesions to extensive macules, addressing fundamental limitations of both traditional SAM architectures and standard Transformer models.

Experimental validation demonstrates that sparse attention mechanisms, MoE design, and enhanced ASPP modules contribute both individually and synergistically to improved segmentation accuracy without computational overhead. Comprehensive evaluation through quantitative benchmarks, ablation studies, and cross-dataset validation confirms state-of-the-art performance with robust generalization across diverse agricultural scenarios.

## 2. Results

### 2.1. Dataset

The primary dataset used for evaluating the proposed Sparse-MoE-SAM framework is the PlantVillage Extended dataset, which has been widely adopted in plant pathology research due to its broad taxonomic coverage and high-quality annotations. The dataset comprises 87,848 RGB images of individual leaves, representing 38 distinct disease categories across various plant species, including tomato, potato, grape, and corn. All images were standardized to 256×256 pixel resolution using bicubic interpolation to ensure consistent input dimensions across batches. Each image is paired with a pixel-level binary segmentation mask $M\in {\left\{0,1\right\}}^{256\times 256}$, where $M_{i,j}=1$ indicates the presence of disease symptoms at pixel position (*i,j*). The dataset was partitioned into training (70%), validation (15%), and testing (15%) subsets using stratified sampling to maintain class distribution proportions. A quantitative summary of the disease class distribution entropy $H_{c}$ was computed using Shannon’s entropy formula, expressed as follows:(1)$$H_{c}=-\sum_{i=1}^{N_{c}}p_{i}\log p_{i}\;\mathrm{where}\;p_{i}=\frac{n_{i}}{\sum_{j}n_{j}}$$
where $n_{i}$ denotes the number of samples for class *i*, and $N_{c}=38$ is the total number of disease categories. The dataset demonstrates moderate class imbalance with an average per-class entropy of $H_{c}=3.74$, which is addressed during training using class-weighted loss.

To assess cross-domain robustness and fine-grained segmentation accuracy under challenging visual conditions, we employ the CVPPP Leaf Segmentation dataset comprising 4477 RGB images of rosette plants captured under controlled greenhouse conditions. Unlike PlantVillage, this dataset focuses on leaf-level segmentation with densely clustered, frequently occluded leaves, making it ideal for evaluating boundary preservation and instance separation. Each image contains instance-level ground truth masks ${\left\{M_{k}\right\}}_{k=1}^{N_{l}}$, where $N_{l}$ denotes the number of leaves and $M_{k}\in {\left\{0,1\right\}}^{H\;\times \;W}$. To generate compatible binary disease masks for training, artificial lesions were synthesized using stochastic lesion functions $L\left(x,y\right)∼N\left(\mu ,\sigma^{2}\right)$, injected at randomized biologically plausible locations to simulate natural disease progression. Segmentation quality was quantified using instance-aware Intersection-over-Union (${\mathrm{IoU}}_{inst}$) and average Hausdorff distance (${\mathrm{HD}}_{avg}$) across test instances. The occlusion complexity κ was measured as follows:(2)$$\kappa =\frac{1}{N_{l}}\sum_{k=1}^{N_{l}}\frac{\mathrm{Area}(M_{k}\cap {⋃}_{j\neq k}M_{j})}{\mathrm{Area}(M_{k})}$$

This metric captures the degree of overlap among leaves, providing an analytical basis for benchmarking segmentation robustness under occlusion-heavy scenarios. To benchmark real-world adaptability, we compiled a custom agricultural field dataset containing 12,340 RGB images acquired from farmland across European and Southeast Asian climate zones. Images were captured using drone-mounted cameras and handheld smartphones under variable photometric conditions including direct sunlight, overcast skies, and nighttime illumination [20]. This dataset provides significant environmental diversity, with recorded photometric variation ($\sigma_{L}$), background clutter index ($C_{b}$), and reflectance noise (η) per image. Plant pathology experts manually annotated disease regions to generate binary masks $M\in {\left\{0,1\right\}}^{H\;\times \;W}$. Quality assurance via inter-annotator agreement analysis yielded a mean Cohen’s kappa score of κ=0.89. Environmental variations were normalized using a domain-adaptive transformation:(3)$$I_{norm}\left(x,y\right)=\frac{I\left(x,y\right)-\mu_{env}}{\sigma_{env}+\epsilon }$$
with $\mu_{env},\sigma_{env}$ estimated from ambient light histograms.

This procedure ensures stable input distribution across varying capture settings, enabling more reliable feature learning under naturalistic agricultural conditions.

Table 1 provides a detailed comparison of the three datasets, highlighting their statistical characteristics and structural differences across key dimensions.

Collectively, these datasets provide a robust evaluation framework for segmentation accuracy and generalization capacity under heterogeneous conditions. Variations in annotation structure, visual complexity, and domain-specific characteristics establish a challenging evaluation benchmark for the proposed segmentation framework. By leveraging structured (PlantVillage), occlusion-intensive controlled (CVPPP), and real-world field-collected datasets, we enable comprehensive assessment of the model’s practical performance capabilities.

To maintain input consistency and training stability, we implemented rigorous image standardization. All input images were resized to 256 × 256 pixels using bicubic interpolation, which preserves local gradients more effectively than bilinear or nearest-neighbor approaches. The resizing transformation is defined by the following function:(4)$$I^{\prime }=R_{256\times 256}^{\mathrm{bicubic}}\left(I\right)$$
where I is the original image and I′ is the transformed image. Following resizing, pixel intensities were normalized using ImageNet statistics to align with the pretrained backbone feature distribution, applying the following:(5)$$I^{\prime \prime }\left(x,y,c\right)=\frac{I^{\prime }\left(x,y,c\right)-\mu_{c}}{\sigma_{c}}$$
where $\mu_{c}$ and $\sigma_{c}$ denote the mean and standard deviation for channel c∈{R,G,B}, respectively. For the segmentation masks, we converted all annotated regions into binary format, where the pixel value is set to 1 if it falls within a disease-infected region, and 0 otherwise. Mathematically, the binary mask transformation is defined as follows:(6)$$M_{bin}\left(x,y\right)=\left\{\begin{array}{cc} 1, & \mathrm{if}\;M\left(x,y\right)\in L_{disease}\\ 0, & \mathrm{otherwise} \end{array}\right.$$
where $L_{disease}$ denotes the set of disease labels and M(x,y) is the original annotated label at pixel location (x,y).

The data cleaning and quality control pipeline consisted of multiple stages to ensure training and evaluation dataset integrity. First, a stratified sampling approach selected 10% of all data across disease classes for manual inspection. Trained annotators reviewed this subset using a standardized rubric assessing segmentation accuracy and annotation completeness. Annotation quality was validated through inter-annotator agreement analysis using Cohen’s Kappa coefficient (κ) as the reliability metric. Let $P_{o}$ represent observed agreement and $P_{e}$ denote chance-expected agreement as follows:(7)$$\kappa =\frac{P_{o}-P_{e}}{1-P_{e}}$$
where values κ>0.85 were considered acceptable, indicating substantial agreement among annotators. Images failing to meet this agreement threshold were flagged for reannotation or removal. Furthermore, corrupted images—defined as those with unreadable pixel arrays or compression artifacts—were filtered using a perceptual hash function and image entropy check. The entropy H of an image I was computed as:(8)$$H\left(I\right)=-\sum_{i=0}^{255}p_{i}\log_{2}\left(p_{i}\right)$$
where $p_{i}$ represents the probability distribution of grayscale pixel intensity *i*; images with entropy H<2.0 were considered low-information and discarded.

To further quantify dataset consistency, we introduced a statistical quality control matrix that assesses intra-class variance across normalized histograms, texture statistics, and structural similarity indices (SSIM). The class-wise histogram divergence was measured using Jensen–Shannon divergence $D_{JS}$ between class centroids $P_{i}$ and image histograms $Q_{i}$ as follows:(9)$$D_{JS}\left(P_{i}\|Q_{i}\right)=\frac{1}{2}D_{KL}\left(P_{i}\|M\right)+\frac{1}{2}D_{KL}\left(Q_{i}\|M\right)$$
where $M=\frac{1}{2}\left(P_{i}+Q_{i}\right)$. For texture consistency analysis, Gray-Level Co-occurrence Matrix (GLCM) metrics including contrast, correlation, and entropy were computed. Outliers exceeding ±3σ beyond class means were flagged. Structural Similarity Index Measure (SSIM) scores were computed pairwise within each disease class and averaged to quantify structural pattern coherence. All metrics were documented in a tabular quality report.

After preprocessing, the filtered dataset contained 85,763 validated samples with homogeneous input distributions across training (70%), validation (15%), and testing (15%) partitions. The final dataset was augmented with metadata descriptors including image entropy, histogram distance to class centroid, and expert confidence score. These were encoded in structured JSON format to support metadata-aware training and reproducibility auditing. Consistent preprocessing across PlantVillage, CVPPP, and field datasets minimized evaluation biases from procedural variations. This quantitatively validated pipeline ensures reliable model training and reproducible cross-domain evaluation in agricultural contexts.

### 2.2. Evaluation Metrics

To rigorously assess the segmentation quality of plant disease regions, we employ a suite of performance metrics widely adopted in semantic segmentation benchmarks. The primary metric is Intersection over Union (IoU), which quantifies the pixel-wise overlap between the predicted segmentation mask and the ground truth. Given a set of predicted pixels P and ground truth pixels G, the IoU is defined as follows:(10)$$\mathrm{IoU}=\frac{\left|P\cap G\right|}{\left|P\cup G\right|}=\frac{TP}{TP+FP+FN}$$
where *TP*, *FP*, and *FN* represent the number of true positives, false positives, and false negatives, respectively. The IoU is particularly sensitive to under-segmentation and over-segmentation, making it crucial for applications where accurate disease area delineation is essential for downstream treatment decisions.

Complementing IoU, the Dice coefficient offers an F1-like evaluation of segmentation performance, emphasizing the balance between precision and recall. Mathematically, the Dice coefficient is given by the following:(11)$$\mathrm{Dice}=\frac{2|P\cap G|}{\left|P|+|G\right|}=\frac{2TP}{2TP+FP+FN}$$

Dice is especially robust to small object instances, which is advantageous in our task where early-stage disease symptoms may occupy minimal spatial regions. Moreover, unlike IoU, Dice score tends to be more lenient in overlapping predictions, which helps evaluate the network’s tolerance to slight boundary deviations.

Precision and recall are also analyzed to measure the model’s classification quality on a per-pixel basis. Precision (P) is defined as follows:(12)$$\mathrm{Precision}=\frac{TP}{TP+FP}$$
and recall (R) as follows:(13)$$\mathrm{Recall}=\frac{TP}{TP+FN}$$

While precision quantifies the model’s ability to avoid false positives such as misclassifying healthy tissue as diseased, recall measures its capability to detect all pathological regions. Plant pathology applications frequently emphasize high recall to prevent overlooking potentially harmful symptoms, though this requires balancing against false positives that might prompt unnecessary treatments.

For boundary accuracy evaluation, the Hausdorff distance (HD) quantifies maximum contour deviation between predicted and ground-truth boundaries. Given two sets of points *A* and *B* representing the predicted and ground truth boundaries, the directed Hausdorff Distance is defined as follows:(14)$$H\left(A,B\right)=\max_{a\in A}\min_{b\in B}||a-{b||}_{2}$$

Hausdorff distance captures worst-case boundary error, making it essential for segmentation models in precision agriculture applications such as targeted pesticide application or robotic treatment systems. Lower HD values indicate superior boundary alignment, enabling accurate quantification of disease severity and spatial spread.

To complement accuracy metrics, computational efficiency was evaluated—a critical factor for real-time deployment in agricultural contexts. Floating Point Operations per Second (FLOPs) measure computational complexity through the number of multiply–add operations per forward pass. The theoretical FLOPs for a convolutional layer is calculated as follows:(15)$${\mathrm{FLOPs}}_{conv}=2\cdot H_{o}\cdot W_{o}\cdot C_{o}\cdot \left(K^{2}\cdot C_{i}\right)$$
where $H_{o},W_{o}$ are the output height and width, $C_{o}$ is output channels, K is the kernel size, and $C_{i}$ is the number of input channels. In our model, sparse attention reduces attention-related FLOPs by replacing dense matrix operations with top-k selective computation, reducing theoretical complexity from O($n^{2}d$) to O(nkd) where k≪n.

Model size was quantified via total learnable parameters (millions) and peak GPU memory consumption (GB) during inference. These metrics determine deployability on edge devices including agricultural drones, smartphones, and robotic platforms. Our mobile variant demonstrates significant reductions in both memory footprint and parameter count without significant degradation in segmentation accuracy.

Real-time performance was evaluated using: inference time (in milliseconds per image), throughput (frames per second, FPS), and energy consumption (in millijoules per inference). These metrics are benchmarked under controlled environments using NVIDIA RTX 3090 GPUs and PyTorch’s built-in profiler. Energy consumption is estimated using GPU power profiles over multiple inference iterations, normalized by batch size and image resolution.

The inference latency T is computed as follows:(16)$$T=\frac{1}{N}\sum_{i=1}^{N}t_{i}$$
where $t_{i}$ is the inference time of the $i^{th}$ sample, and throughput is given by the following:(17)$$FPS=\frac{1}{T}$$

Energy consumption per image is expressed as follows:(18)E=P·T
where *P* is average power draw (W) during inference. These metrics provide a comprehensive perspective on real-world feasibility, especially under energy-constrained field deployment conditions.

### 2.3. Hardware and Software Environment

As illustrated in Table 2. The Sparse-MoE-SAM framework is deployed on a dedicated deep learning workstation optimized for multi-branch architectures and memory-intensive operations. Hardware specifications include the following: GPU: NVIDIA RTX 3090 (24GB GDDR6X VRAM) to support batched expert routing and sparse attention during backpropagation. CPU: Intel Core i9-11900K (8 cores/16 threads @ 3.5 GHz base) for efficient data loading/preprocessing. RAM: 64GB DDR4 @ 3200MHz to alleviate memory constraints during high-throughput augmentation.

The software stack is built on PyTorch (v1.13.0), selected for its flexibility in implementing custom attention masks and expert routing logic. CUDA 11.7 provides native GPU acceleration, leveraging Tensor Cores to significantly accelerate matrix operations in attention and expert modules. Python 3.9.7 serves as the core programming language for model definition, training loops, and evaluation scripts. We adopt the Apex library for mixed-precision training, Albumentations for advanced data augmentation, and TorchMetrics for standardized metric computation [21]. The entire development environment is containerized using Docker to ensure reproducibility and cross-system compatibility.

The implementation leverages PyTorch’s dynamic computation graph capabilities to integrate sparse attention and MoE without modifying the backpropagation pipeline. Sparse attention masks are constructed using top-k selection via PyTorch’s torch.topk() function, followed by zero-masking all non-selected attention scores. The MoE module is implemented using gated routing with token-level dispatch. Expert layers are constructed as parallel convolutional or transformer sub-networks, dynamically selected per token batch using scatter_add operations for efficient gradient accumulation. This dynamic routing strategy necessitates customized autograd functions for memory-efficient backward propagation.

Performance profiling uses PyTorch Profiler and NVIDIA Nsight to measure GPU memory footprint, FLOPs, and kernel execution time. These tools confirm that sparse attention reduces average memory usage by ∼34% compared to dense attention. Furthermore, the MoE module incurs negligible routing overhead (<7% runtime increase versus standard feedforward layers) due to parallel expert computation. End-to-end training (including all ablation variants) is orchestrated via PyTorch Lightning for structured experiment tracking, reproducibility, and seamless TensorBoard integration.

The following is an overview of the hardware/software stack:

**Table 2 plants-14-02634-t002:** Experimental hardware and software configuration.

Environment Component	Specification	Details
GPU	NVIDIA RTX 3090	24GB GDDR6X, CUDA cores optimized for tensor ops
CPU	Intel Core i9-11900K	8-core, 16-thread, 3.5 GHz
RAM	64 GB DDR4	3200 MHz, multi-channel memory
OS	Ubuntu 20.04 LTS	Docker containerized execution
Framework	PyTorch 1.13.0	With CUDA 11.7, cuDNN enabled
Mixed Precision	Apex AMP	Reduces memory usage, accelerates forward/backward pass
Training Management	PyTorch Lightning	Modular experiment design and logging
Profiling	PyTorch Profiler, Nsight Systems	FLOPs, kernel time, memory bandwidth

### 2.4. Segmentation Performance Comparison

As illustrated in Table 3. Quantitative evaluation on the PlantVillage Extended dataset confirms the superior performance of our proposed approach across multiple metrics. The comprehensive comparison includes traditional segmentation architectures, modern transformer-based methods (SegFormer), and foundation models. The baseline U-Net achieves 86.3% IoU, 89.7% Dice, 88.4% precision, and 87.2% recall with 31.0 M parameters and 15.2 GFLOPs. Attention U-Net shows modest improvements at 88.1% IoU and 91.2% Dice with 34.9 M parameters and 18.7 GFLOPs. DeepLabV3+ achieves stronger performance (89.4% IoU, 92.1% Dice) using 41.3 M parameters and 22.8 GFLOPs. The transformer-based SegFormer reaches 90.2% IoU and 92.8% Dice but requires 64.1 M parameters and 31.5 GFLOPs.

Boundary accuracy assessment via Hausdorff Distance reveals significant contour precision improvements. Our full model achieves the lowest Hausdorff Distance (1.87 mm), representing 22.4% and 31.5% improvements over Standard SAM (2.41 mm) and U-Net (3.42 mm), respectively. This reduction demonstrates our sparse attention mechanism’s effectiveness in preserving sharp disease boundaries—critical for treatment planning and severity assessment. Progressive improvements from CNN-based to transformer-based methods highlight global context modeling importance, while our sparse refinement further enhances boundary delineation without computational overhead. The mobile variant maintains competitive accuracy (2.15 mm), confirming architectural optimizations preserve spatial precision.

As illustrated in Table 4. Cross-dataset performance analysis reveals robust generalization capabilities across diverse imaging conditions and data distributions. While all methods exhibit reduced performance when transitioning from controlled laboratory conditions (PlantVillage) to complex real-world scenarios (CVPPP and Custom Field), our approach maintains the smallest IoU performance gap. The 4.5-percentage-point decline from PlantVillage (94.2% IoU) to Custom Field (87.4% IoU) compares favorably to SAM’s 6.9-point decline and SegFormer’s 7.0-point reduction. This superior domain adaptation stems from the biologically motivated sparse attention patterns and adaptive specialization of expert modules, which intrinsically adapt to diverse symptom manifestations across imaging environments.

Complex Leaf Occlusion Performance Analysis: The CVPPP dataset provides an ideal testbed for evaluating performance under severe leaf occlusion conditions, with overlap ratios ranging from 15% to 85%. Our method achieved 89.7% IoU on CVPPP, demonstrating exceptional capability in handling overlapping structures. We conducted detailed analysis across three occlusion severity levels: mild (15–35% overlap), moderate (35–60% overlap), and severe (60–85% overlap). Under mild occlusion, our method maintains 92.3% IoU with only 1.9% degradation from isolated leaf scenarios. For moderate occlusion, IoU drops to 88.1% (6.1% degradation), while severe occlusion results in 84.2% IoU (10.0% degradation). Importantly, our dual-stage MoE decoder excels in these challenging scenarios through specialized expert routing: the first-stage experts focus on global context to identify partially visible leaf boundaries, while second-stage experts emphasize edge refinement and boundary disambiguation. The sparse attention mechanism particularly benefits occlusion handling by concentrating computational resources on visible leaf regions rather than occluded areas, leading to more accurate boundary delineation. Comparative analysis shows our method outperforms Standard SAM by 4.3% IoU under severe occlusion conditions, demonstrating the effectiveness of expert specialization for complex spatial reasoning tasks.

Low-Light Condition Performance: Agricultural imaging often occurs under suboptimal lighting conditions, including dawn/dusk field work, indoor greenhouse settings, and cloudy weather. We evaluated model robustness across synthetic low-light scenarios by systematically reducing image brightness and adding realistic noise patterns. Images were processed with brightness reduction factors of 0.7, 0.5, and 0.3 (representing moderate, low, and very low light conditions), combined with Gaussian noise (σ=0.05) to simulate sensor noise under high ISO settings. Under moderate low-light conditions (0.7× brightness), our method maintains 91.8% IoU with only 2.4% degradation from normal lighting. Performance drops to 89.3% IoU (4.9% degradation) under low-light (0.5× brightness) and 85.7% IoU (8.5% degradation) under very low-light conditions (0.3× brightness). The sparse attention mechanism proves particularly valuable in these scenarios by concentrating computational resources on the most informative regions, effectively filtering out noise-dominated background areas. Our dual-stage MoE architecture adapts to varying lighting through expert specialization: certain experts become specialized for contrast enhancement while others focus on edge preservation under noise. Compared to Standard SAM, our method shows superior robustness with 3.1% better IoU under very low-light conditions, validating the framework’s applicability for practical agricultural deployment across diverse environmental conditions.

Analysis of Challenging Disease Types:While our method demonstrates consistent improvements across disease categories, certain pathological conditions present greater segmentation challenges. Mosaic Virus, despite showing the highest improvement margin, achieves the lowest absolute IoU (93.2%) among the evaluated diseases. This difficulty stems from viral symptoms manifesting as subtle color variations and irregular mottled patterns that lack clear boundaries, making precise segmentation inherently challenging. The sparse attention mechanism, while generally beneficial, occasionally struggles with such diffuse symptomatology where the entire leaf surface may be affected rather than discrete lesional areas. Similarly, Late Blight presents challenges due to its rapidly evolving nature, where symptoms progress from small water-soaked spots to large irregular patches with indistinct borders. Our analysis reveals that diseases with diffuse symptom patterns (Mosaic Virus: 93.2% IoU), rapidly changing morphology (Late Blight: 93.8% IoU), and subtle early-stage manifestations achieve lower performance compared to diseases with distinct, well-defined lesions like Rust (95.4% IoU) and Bacterial Spot (95.1% IoU). Future work should address these limitations through specialized expert training for handling gradual color transitions and incorporating temporal information for diseases with dynamic progression patterns.

Consistent improvement margins across disease types confirm the generalizability of our architectural innovations, indicating that learned specializations adapt effectively to varying pathological characteristics while maintaining specificity.

As illustrated in Figure 1. Foundation model comparison indicates Standard SAM attains 91.7% IoU, 94.2% Dice, 93.1% precision, and 92.4% recall with substantial computational requirements (636.0 M parameters, 187.3 GFLOPs). FastSAM provides a more efficient alternative (90.8% IoU, 93.5% Dice) using 68.0 M parameters and 45.2 GFLOPs. Our proposed full-model achieves superior segmentation performance: 94.2% IoU, 96.1% Dice, 95.3% precision, and 94.8% recall with only 142.7 M parameters and 142.9 GFLOPs. This configuration yields a 23.7% computational cost reduction versus Standard SAM while improving IoU by 2.5 percentage points. The mobile variant maintains competitive performance (92.1% IoU, 94.7% Dice) with dramatic resource reduction (45.3 M parameters, 38.7 GFLOPs), representing 77.5% fewer parameters than the full-model without significant performance degradation.

### 2.5. Computational Efficiency Analysis

The proposed method exhibits superior computational efficiency-segmentation performance trade-offs across evaluation dimensions. Comprehensive analysis confirms a 23.7% reduction in FLOPs with a simultaneous 2.5% IoU improvement versus baseline SAM. Memory efficiency analysis reveals 35% lower peak memory usage during inference, enabling deployment on resource-constrained agricultural devices with limited VRAM. Inference speed evaluation shows 2.3× faster processing than standard SAM (102 ms vs. 234 ms per 256 × 256 image), facilitating real-time field deployment for agricultural monitoring. Computational complexity analysis confirms sparse attention’s theoretical advantages: dense attention exhibits prohibitive quadratic scaling for high-resolution imagery, while our sparse approach maintains linear scaling.

As illustrated in Figure 2. Memory usage comparison shows our method requires 6.7 GB peak memory versus SAM’s 15.8 GB, enabling deployment on consumer-grade agricultural research GPUs. Table 5 details practical deployment characteristics across segmentation methods. Traditional CNN-based methods (e.g., U-Net) consume only 2.1 GB but achieve limited accuracy, whereas foundation models (e.g., SAM) require 15.8 GB. Our approach balances this trade-off, requiring 6.7 GB (full model) with state-of-the-art performance.

To evaluate real-world deployment feasibility, we conducted extensive performance testing on resource-constrained devices commonly used in agricultural settings. On a NVIDIA Jetson Xavier NX (6 GB RAM, 384 CUDA cores), our mobile variant achieves 92.1% IoU with 3.2 s inference time per 256 × 256 image, compared to standard SAM which fails to run due to memory constraints. The mobile variant successfully processes images on a Jetson Nano (4 GB RAM) with 4.8 s inference time while maintaining 91.3% IoU. On smartphone-grade hardware (Snapdragon 855 with Adreno 640 GPU), our optimized model achieves 89.7% IoU with 6.1 s processing time using TensorFlow Lite optimization. These results demonstrate practical viability for edge deployment in precision agriculture applications where real-time processing is less critical than accuracy and device compatibility.

To reduce the model’s computational load, we apply pruning techniques that reduce the number of parameters and operations while maintaining accuracy. Specifically, sparse attention mechanisms are employed, which allow the model to focus only on the most relevant features, significantly lowering the model size. For mobile deployment, we convert the model into TensorFlow Lite format or PyTorch Mobile, both of which optimize the model for inference on edge devices. These tools provide optimized kernels for mobile GPUs, ensuring efficient resource use. Smartphones or drones equipped with the optimized model can be used by field scouts for rapid disease identification during routine inspections. The model processes the leaf images locally, offering instant disease detection without requiring internet connectivity.

Mobile Device Application Scenarios: Our framework enables diverse practical deployment scenarios across agricultural value chains. Scenario 1—Field Scout Applications: Agricultural scouts equipped with smartphones can capture leaf images during routine field inspections. The mobile variant processes images locally within 6.1 s, providing immediate disease identification without requiring internet connectivity. This enables rapid decision-making for treatment timing and resource allocation in remote farming areas. Scenario 2—Farmer Decision Support: Small-scale farmers using entry-level smartphones can photograph suspected disease symptoms for on-device analysis. The compressed model provides actionable recommendations about disease severity and treatment urgency, supporting precision application of pesticides and reducing unnecessary chemical usage. Scenario 3—Agricultural Extension Services: Extension agents can use tablet devices during farmer training sessions, demonstrating disease identification in real-time and building local diagnostic capacity. The mobile framework processes multiple leaf samples during field workshops, facilitating hands-on learning experiences. Scenario 4—Supply Chain Quality Control: Post-harvest facilities can deploy our framework on mobile devices for quality assessment of incoming produce, automatically flagging potentially diseased materials before storage or distribution. Processing times of 4–6 s per image enable integration into existing quality control workflows without significant throughput impacts.

The mobile variant achieves exceptional 3.1 GB memory efficiency (80.4% reduction vs. SAM), enabling edge deployment under 4–8 GB GPU constraints. Inference time analysis shows our mobile variant processes images in 58 ms (4.0× faster than SAM’s 234 ms) on high-end hardware. Throughput reaches 17.2 frames/s, exceeding the 15 FPS threshold for smooth real-time processing while maintaining competitive accuracy. Energy consumption analysis reveals our mobile variant uses 189 mJ/inference versus SAM’s 1024 mJ, enabling extended operation in battery-powered field deployments. All variants exhibit speed-up improvements, with our full model achieving 2.3× acceleration over SAM alongside superior accuracy, validating sparse attention and expert routing optimizations.

### 2.6. Expert Utilization Analysis

Expert selection frequency analysis reveals distinct specialization patterns with balanced utilization across processing stages. During Stage 1 (coarse segmentation), standard convolution experts account for 28.4% of selections, demonstrating effectiveness in general pattern recognition. Dilated convolution experts achieve highest utilization (31.7%), reflecting superior capability for large disease regions and contextual relationships across extended spatial neighborhoods. Depthwise separable convolution experts maintain 25.9% utilization, while specialized context-aware experts contribute 14.0% for complex spatial reasoning.

Stage 2 refinement exhibits specialized adaptation: Standard convolution utilization increases to 35.2%, highlighting their role in boundary refinement. Dilated convolution decreases to 23.1% as large-scale contextual processing diminishes. Depthwise separable convolution achieves peak utilization (41.7%), confirming efficiency advantages for fine-grained boundary delineation. Balanced utilization prevents expert collapse, and stage-specific specialization emerges through training without explicit supervision, validating the gating mechanism’s effectiveness.

Load balancing analysis demonstrates the auxiliary balance loss effectiveness in maintaining expert diversity during training. Expert entropy increases from 1.89 (initial) to 1.99 (final), approaching the theoretical maximum ($\log_{2}$(4) = 2.0) for four experts, indicating near-optimal load distribution. The Gini coefficient decreases from 0.42 to 0.33, reflecting improved utilization equality [18]. Load variance reduces from 0.087 to 0.058, mitigating expert utilization skewness. Balance loss consistently declines from 0.034 to 0.019, confirming the gating network learns balanced routing without explicit supervision. Crucially, expert collapse remains at 0% throughout training, confirming our mechanism prevents the single-expert degradation characteristic of mixture-of-experts architectures in specialized domains.

As illustrated in Table 6. Attention diversity and sparsity analysis reveal specialized patterns across the eight heads of our sparse attention mechanism. Sparsity ratios consistently exceed 88.9% (average: 90.5%), confirming effective pruning of irrelevant connections while preserving critical spatial relationships. Head 5 achieves peak sparsity (92.1%) when processing texture variations, indicating selective attention for textural disease symptoms. Entropy scores range from 2.06 ti 2.19 (average: 2.11), demonstrating balanced attention distribution within retained connections.

Specialization analysis shows biologically aligned focus: Head 2 exhibits strongest pathologist attention correlation (0.87) for necrotic regions, validating detection of advanced symptoms. Head 1 effectively captures fine lesions (0.84 correlation), crucial for early detection. Head 7 shows 0.86 correlation for disease progression patterns, indicating temporal development identification. Head 8’s lower background correlation (0.78) reflects reduced alignment need. The average 0.83 correlation confirms our sparse attention learns biologically relevant representations, reconciling computational efficiency with plant pathology expertise.

### 2.7. Ablation Study Results

As illustrated in Table 7. Comprehensive ablation study systematically evaluates each component’s contribution. Baseline SAM establishes foundation performance (91.7% IoU, 187.3 GFLOPs). Incorporating sparse attention improves IoU to 92.4% while reducing computation to 156.2 GFLOPs (0.7-pp IoU gain, 16.6% FLOPs reduction), confirming sparse attention maintains quality with strategic pruning.

Adding first-stage Mixture of Experts (MoE) elevates IoU to 93.1% at 148.7 GFLOPs (cumulative 1.4-pp IoU gain, 20.6% FLOPs reduction versus baseline). Enhanced ASPP module integration boosts IoU to 93.6% (145.3 GFLOPs), validating its multi-scale processing value. Final second-stage MoE decoder achieves 94.2% IoU and 142.9 GFLOPs (cumulative 2.5-pp IoU gain, 23.7% FLOPs reduction). Progressive improvements demonstrate meaningful contributions: sparse attention enables greatest computation savings, while each MoE stage adds ≈0.7-pp IoU through task-specialized feature processing.

Statistical Significance Analysis: To validate the reliability of our ablation results, we conducted statistical significance testing across five independent training runs with different random seeds. Using paired *t*-tests with Bonferroni correction for multiple comparisons, we found all IoU improvements to be statistically significant at *p* < 0.01 level. Specifically: sparse attention addition (92.4% vs. 91.7%, *p* = 0.003), first-stage MoE integration (93.1% vs. 92.4%, *p* = 0.007), enhanced ASPP incorporation (93.6% vs. 93.1%, *p* = 0.009), and final second-stage MoE (94.2% vs. 93.6%, *p* = 0.004). The confidence intervals (95% CI) for each improvement are: sparse attention [0.4%, 1.0%], first-stage MoE [0.4%, 1.0%], enhanced ASPP [0.2%, 0.8%], and second-stage MoE [0.3%, 0.9%]. These results confirm that observed performance gains are not due to random variation and represent genuine architectural contributions. Furthermore, ANOVA analysis (F(4,20) = 127.3, *p* < 0.001) confirms significant differences between configurations, with effect size ($\eta^{2}$ = 0.962) indicating that architectural choices explain 96.2% of performance variance.

Extended Parameter Sensitivity Analysis: To provide comprehensive understanding of design choices, we conducted extensive parameter sensitivity studies. Sparse Attention Top-K Analysis: We systematically varied the top-k retention ratio ρ from 0.05 to 0.3. Results show optimal performance at ρ = 0.1 (94.2% IoU), with degradation at both extremes: ρ = 0.05 achieves 92.1% IoU due to information loss, while ρ = 0.3 reaches 93.4% IoU with increased computational cost. The sweet spot at ρ = 0.1 balances accuracy retention with 23.7% FLOPs reduction.

MoE Expert Count Variation: We evaluated expert numbers from 2 to 8 for both stages. Stage 1 performs optimally with 4 experts (94.2% IoU); fewer experts (2–3) achieve 92.8–93.5% IoU due to insufficient specialization, while more experts (6–8) show marginal improvement (94.0–94.1% IoU) with significantly increased parameters. Stage 2 benefits from 3 experts (94.2% IoU) versus 2 experts (93.7% IoU) or 4+ experts (94.0–94.1% IoU with parameter overhead).

Gating Network Architecture: We tested gating depths from 1 to 4 layers. Single-layer gating achieves 93.1% IoU with rapid routing decisions but limited capacity for complex feature analysis. Two-layer gating (our choice) reaches 94.2% IoU, optimally balancing routing sophistication with computational efficiency. Deeper networks (3–4 layers) provide minimal gains (94.0–94.1% IoU) while increasing overhead.

Loss Function Weight Sensitivity: Balance loss weight $\lambda_{1}$ analysis shows optimal performance at 0.01; lower values (0.001–0.005) lead to expert collapse (91.8–92.6% IoU), while higher values (0.05–0.1) over-regularize expert diversity (92.4–93.1% IoU). Sparsity weight $\lambda_{2}$ optimal at 0.001 balances attention focus with flexibility.

### 2.8. Case Study

As illustrated in Figure 3, comprehensive visualization analysis reveals sophisticated internal mechanisms of our Sparse-MoE-SAM framework through attention pattern examination and expert activation mapping.

Attention Map Visualization: Sparse attention visualization shows focused activation patterns (peak weights: 0.85–0.94) precisely aligned with disease-affected regions, with minimal activation (<0.08) in healthy tissues. Figure 3A displays attention heatmaps overlaid on original images for representative disease cases: bacterial spot detection shows concentrated attention on circular lesions with clear boundaries, late blight attention maps highlight water-soaked margins characteristic of fungal infections, and mosaic virus attention patterns reveal the model’s ability to detect subtle color variations across leaf surfaces. Cross-validation with expert pathologist annotations confirms strong spatial alignment (Pearson correlation >0.84 across disease types). Multi-head attention analysis demonstrates specialized preferences: Head 1 focuses on textural changes (entropy-based features), Head 2 emphasizes color transitions (HSV space variations), Heads 3–4 capture boundary characteristics (gradient-based features), and Heads 5–8 integrate multi-scale contextual information.

Expert Routing Visualization: Figure 3B illustrates dynamic expert activation patterns through routing probability heatmaps. For bacterial spot cases, dilated convolution experts show 73% activation in central lesion areas requiring broad contextual analysis, while standard convolution experts dominate edge regions (68% activation) for precise boundary delineation. Depthwise separable experts preferentially activate in computationally constrained scenarios (mobile deployment), achieving 81% routing efficiency. Temporal stability analysis across video sequences shows consistent expert selection patterns (correlation coefficient >0.79), indicating robust specializations that generalize across samples while adapting to local feature complexity. Stage-wise routing analysis reveals progressive specialization: Stage 1 achieves coarse disease localization with balanced expert utilization (25–35% per expert), while Stage 2 shows task-specific expert dominance based on lesion characteristics.

Expert activation heatmaps illustrate adaptive routing: dilated convolution experts preferentially activate in necrotic regions requiring broad context, while depthwise separable experts show increased utilization in efficiency-critical areas. Temporal consistency analysis reveals stable expert selection patterns (>0.79 correlation coefficient), indicating robust specializations generalizing across samples while sensitive to local feature complexity.

Qualitative segmentation results demonstrate superior boundary accuracy and reduced false positives (3.2%) in challenging agricultural scenarios. Complex background discrimination achieves 93.8% boundary accuracy against environmental interference (soil, shadows, leaf artifacts). Dual-stage MoE refinement capability is evident in overlapping symptom scenarios: Stage 1 establishes 89.4% boundary accuracy, Stage 2 refines to 91.7% through specialized routing. Early-stage lesion detection shows 90.9% accuracy for sub-15-pixel lesions, outperforming human perception. Variable illumination analysis maintains >89.6% boundary accuracy across diverse lighting, color temperature, and shadow conditions. These results validate that our architecture enhances computational efficiency, biological relevance, and clinical utility, establishing new standards for interpretable plant disease segmentation.

## 3. Discussion

This research addresses deployment challenges for high-precision plant disease segmentation in agricultural settings where computational resources are limited. Field conditions introduce complex backgrounds, variable lighting, and sparse disease symptoms that challenge traditional dense attention models.

Prior studies predominantly employ dense attention frameworks (e.g., ViT, SAM), effective for general segmentation but exhibiting quadratic computational complexity that hinders real-time edge deployment. We introduce Sparse-MoE-SAM, integrating sparse attention with a Mixture-of-Experts (MoE) decoding structure to address computational bottlenecks and biological feature sparsity [22,23].

The key novelty of our approach lies in three fundamental departures from existing methods. First, unlike traditional dense attention mechanisms that compute attention weights for all pairwise token interactions, our sparse attention mechanism selectively attends to only the top-k most relevant tokens per query, reducing computational complexity from O(n^2^) to O(nk). This bio-inspired design mirrors how plant pathologists focus attention on disease-relevant regions rather than processing entire leaf surfaces uniformly. Second, our dual-stage MoE decoder employs task-conditional expert routing where different experts specialize in processing distinct morphological patterns (e.g., necrotic spots, chlorotic regions, fungal structures). This contrasts with standard single-decoder architectures that use uniform processing across all spatial regions. Third, our framework integrates adaptive sparsity patterns that dynamically adjust based on disease symptom distribution, unlike fixed sparsity masks used in prior efficient attention methods.

Our domain-adaptive architecture leverages biologically inspired structural priors—particularly the spatial sparsity and multi-scale distribution of disease symptoms. By activating attention only in semantically relevant regions, our model improves mean Intersection over Union (mIoU) by 2.5% while reducing FLOPs by 23.7% compared to SAM. It achieves 2.3× faster inference with 35% lower peak memory usage for 256 × 256 images, enabling deployment on consumer-grade GPUs.

The framework centers on a two-stage MoE decoder that progressively refines boundaries under field conditions [24]. Gate-controlled expert routing allocates computation according to spatial complexity, capturing fine-grained patterns under occlusion or illumination noise. This hierarchical specialization advances pixel-level segmentation beyond dense architectures. Our approach demonstrates strong generalization across datasets (PlantVillage, field-collected data), highlighting practical alignment of architectural design with agricultural domain knowledge.

Several limitations of our study warrant discussion. First, our evaluation is primarily conducted on controlled datasets (PlantVillage) with high-quality annotations, which may not fully represent the complexity of real-world field conditions including soil occlusion, water droplets, and pest damage. Second, the sparse attention mechanism, while computationally efficient, may occasionally miss subtle disease symptoms that manifest as diffuse patterns across large leaf areas. Third, our MoE architecture requires careful hyperparameter tuning for optimal expert utilization, and the learned expert specializations may not transfer effectively across significantly different plant species or disease types. Fourth, the model’s performance on low-resolution imagery (below 128 × 128 pixels) degrades substantially, limiting applicability in scenarios with poor camera quality or distant capture distances. Additionally, computational benefits of sparse attention are most pronounced on modern GPUs with efficient sparse matrix operations; older hardware may not fully realize these advantages. Future work should explore automated expert specialization via neural architecture search, extension to semi-supervised learning regimes to reduce annotation costs, and multimodal sensory data integration (thermal/hyperspectral imaging) to enhance environmental variability handling [25,26].

## 4. Materials and Methods

### 4.1. Overall Framework Architecture

The Sparse-MoE-SAM framework maintains SAM’s encoder generalization while incorporating two domain-specific adaptations: (1) biologically gated sparse attention, and (2) task-conditional MoE architecture [27,28]. These enable efficient, context-aware segmentation with enhanced discriminative capability for lesion regions. As Figure 4 illustrates, our end-to-end system comprises four integrated modules:A sparsity-constrained ViT encoder guided by lesion distribution priors, An adaptive MoE module embedded in the encoder–decoder pathway, A sparse-enhanced ASPP module extracting multi-scale lesion features and A dual-stage decoder with progressive boundary optimization via stage-wise expert routing [29,30]. These components are jointly optimized through a composite loss function that simultaneously reinforces segmentation fidelity, expert utilization balance, and attention sparsity. This alignment ensures internal representations satisfy both computational constraints (O(nk) complexity) and pathological feature semantics required for agricultural deployment.

The Sparse Vision Transformer Encoder modifies SAM’s ViT backbone by replacing dense multi-head self-attention with learnable sparsity patterns. Within each attention head, we dynamically select the *k* most relevant tokens per query position based on input-adaptive importance scores. Given an input token sequence $X\in R^{n\;\times \;d}$, the standard attention mechanism computes as follows:(19)$$\mathrm{Attention}\left(Q,K,V\right)=\mathrm{softmax}\left(\frac{QK^{T}}{\sqrt{d_{k}}}\right)V$$
where $Q=XW^{Q}$, $K=XW^{K}$, and $V=XW^{V}$ are the projected query, key, and value matrices. We redefine this as a top-*k* sparse attention mechanism, where for each query vector $q_{i}$, only the top-*k* keys ${\left\{k_{j}\right\}}_{j\in J_{i}}$ are selected based on the magnitude of their dot-product similarity. The sparsified attention formulation is presented in Equation (Equation 20) as follows:(20)$$\mathrm{SparseAttention}\left(q_{i},K,V\right)=\sum_{j\in J_{i}}\frac{\exp\left(\frac{q_{i}^{T}k_{j}}{\sqrt{d_{k}}}\right)}{\sum_{l\in J_{i}}\exp\left(\frac{q_{i}^{T}k_{l}}{\sqrt{d_{k}}}\right)}v_{j}$$

The sparse attention reduces complexity from $O(n^{2}d)$ to O(nkd), where k=ρn (ρ≪1) and *d* is the feature dimension. Critically, the sparsity mask $I_{i}$ is dynamically regenerated per attention head and per input sample, enabling input-adaptive sparsity patterns aligned with lesion distributions. During training, sparsity is enforced as a hard constraint via Gumbel–Softmax-based differentiable top-*k* approximation.

To enhance feature processing specialization, we insert an MoE module between encoder and decoder stages. It consists of M expert networks ${\left\{f_{j}\right\}}_{j=1}^{M}$, each with distinct inductive biases (e.g., receptive fields, texture sensitivity) optimized for specific spatial/morphological symptom scales. A gating network $G:R^{d}\to R^{M}$ routes input features $F\in R^{h\;\times \;w\;\times \;d}$ by computing routing weights: $\alpha_{i}=\mathrm{softmax}{\left(G\left(F\right)\right)}_{i}$. The final output aggregates expert contributions as specified in Equation (Equation 21) as follows:(21)$$f_{\mathrm{MoE}}\left(F\right)=\sum_{i=1}^{k}\alpha_{i}f_{i}\left(F\right)$$
where k<M controls sparsity at the expert level, and only top-*k* experts are selected per token. To encourage balanced expert usage and prevent expert collapse, we introduce a load balancing loss term, The corresponding expression is provided in Equation (Equation 22) as follows:(22)$$L_{\mathrm{balance}}=\frac{M\cdot \sum_{i=1}^{M}f_{i}^{2}}{{\left(\sum_{i=1}^{M}f_{i}\right)}^{2}}$$
where $f_{i}$ denotes the fraction of tokens routed to expert *i*. The routing network trains jointly with the model, enabling dynamic adaptation to disease regions’ visual complexity and severity.

Subsequent to the encoder, we integrate an Enhanced ASPP module with sparse attention to capture multi-scale context efficiently. This module employs dilated convolutions at rates r∈{1,6,12,18} to generate multi-resolution features $\left\{F_{r}\right\}$, which are concatenated and refined via sparse attention. The attention-enhanced fusion computes as follows:(23)$$F_{\mathrm{fused}}=\mathrm{SparseAttention}\left(\mathrm{Concat}\left(\left[F_{r}\right]\right),\mathrm{Concat}\left(\left[F_{r}\right]\right),\mathrm{Concat}\left(\left[F_{r}\right]\right)\right)$$

This selective refinement operates over concatenated features while maintaining linear complexity. We apply projection layer $W_{p}\in R^{4d\;\times \;d}$ to align fused outputs to the target embedding space, ensuring capture of both fine-grained and large-scale disease cues without excessive computation.

The Dual-Stage MoE Decoder progressively refines segmentation masks through cascaded stages. Stage 1 processes encoder-generated coarse semantic maps to predict initial disease regions. Stage 2 refines boundaries and recovers lesion structures. Denoting stage outputs as $M^{\left(1\right)}\in R^{h\;\times \;w}$ and $M^{\left(2\right)}$. Each decoding stage utilizes its own MoE block, The expressions for $M^{\left(1\right)}$ and $M^{\left(2\right)}$ are given in Equation (Equation 24) as follows:(24)$$M^{\left(1\right)}=\sum_{i=1}^{k}\beta_{i}^{\left(1\right)}D_{i}^{\left(1\right)}\left(F\right),\;M^{\left(2\right)}=\sum_{j=1}^{k}\beta_{j}^{\left(2\right)}D_{j}^{\left(2\right)}\left(\left[F,M^{\left(1\right)}\right]\right)$$
where $D_{i}^{\left(s\right)}$ denotes the *i*-th decoder expert at stage s∈{1,2}, and $\beta_{i}^{\left(s\right)}$ represents top-*k* routing scores from the gating network. The stage-2 decoder concatenates encoder outputs with stage-1 decoder features, enabling hierarchical refinement.

The complete Sparse-MoE-SAM framework optimizes a composite loss function combining segmentation fidelity, attention sparsity, and expert diversity. The objective function is defined as follows:(25)$$L_{\mathrm{total}}=L_{\mathrm{seg}}+\lambda_{1}L_{\mathrm{balance}}+\lambda_{2}L_{\mathrm{sparse}}$$
where $L_{\mathrm{seg}}$ includes binary cross-entropy and Dice coefficient terms, as shown in Equation (Equation 26) as follows:(26)$$L_{\mathrm{seg}}=\mathrm{BCE}\left(M^{\left(2\right)},Y\right)+1-\frac{2\cdot |M^{\left(2\right)}\cap Y|}{|M^{\left(2\right)}\left|+|Y\right|}$$
and $L_{\mathrm{sparse}}$ is an $\ell_{1}$-norm regularization applied to the attention maps, as defined in Equation (Equation 27) as follows:(27)$$L_{\mathrm{sparse}}=\sum_{i,j}\left|\right|A_{i,j}{\left|\right|}_{1}$$
where $A_{i,j}$ denotes attention weights between query *i* and key *j*. This sparsity regularization term enforces localized attention patterns, aligning with plant diseases’ hierarchical visual features. Experimental validation (Section 2.7) confirms this formulation yields quantifiable improvements in segmentation performance and computational efficiency, enhancing suitability for agricultural field settings.

### 4.2. Sparse Attention Mechanism

Traditional vision transformer self-attention mechanisms exhibit O(n2) computational complexity due to dense pairwise token interactions [13]. For an input token sequence $X\in R^{n\;\times \;d}$ where *n* denotes spatial tokens (e.g., image patches) and *d* represents feature dimensionality, standard attention computes as follows:(28)$$\mathrm{Attention}\left(Q,K,V\right)=\mathrm{softmax}\left(\frac{QK^{T}}{\sqrt{d}}\right)V,$$

As illustrated in Figure 5, where $Q=XW^{Q}$, $K=XW^{K}$, and $V=XW^{V}$ are learned linear projections, the resulting n × n attention map incurs prohibitive computational cost for high-resolution images. To address this, we propose sparsity-constrained attention that limits each query $q_{i}$ to attend only to a top-*k* subset of keys $k_{j\in J_{i}}$. Here, $J_{i}$ denotes the indices of the *k* most relevant positions for $q_{i}$ based on dot product similarity as follows:(29)$$\mathrm{SparseAttention}\left(q_{i},K,V\right)=\sum_{j\in J_{i}}\frac{\exp\left(\frac{q_{i}^{T}k_{j}}{\sqrt{d}}\right)}{\sum_{l\in J_{i}}\exp\left(\frac{q_{i}^{T}k_{l}}{\sqrt{d}}\right)}v_{j}.$$

As illustrated in Figure 6, this formulation reduces the attention cost to O(nkd), with k=ρn and ρ≪1. The sparsity mask $J_{i}$ is dynamically determined for each sample and each attention head. To make the selection differentiable, we employ a Gumbel–Softmax-based sampling strategy, approximating top-k through the following:(30)$$J_{i}=\mathrm{top}-k\left(\mathrm{Gumbel}\left(q_{i}^{T}K\right)\right)$$
where Gumbel noise is injected to enable gradient propagation through the argmax-like selection operation.

Gumbel-TopK Strategy Justification: The choice of Gumbel-TopK selection over alternative sparse attention strategies is motivated by several critical advantages for plant disease segmentation. First, differentiability: Standard top-k selection involves discrete argmax operations that break gradient flow during backpropagation. Gumbel–Softmax provides a continuous relaxation that maintains gradient propagation while approximating discrete selection, essential for end-to-end training of our MoE architecture. Second, adaptive exploration: Gumbel noise introduces controlled stochasticity that prevents the attention mechanism from prematurely converging to suboptimal sparse patterns, particularly important given the diverse morphological patterns of plant diseases. Third, biological plausibility: The noise-injected selection mimics the variability in human visual attention when examining diseased leaves, where experts may focus on different but equally relevant symptom regions. Fourth, computational stability: Unlike learned sparse patterns that can collapse during training, Gumbel-TopK maintains consistent sparsity ratios throughout optimization. We empirically validated this choice by comparing against fixed sparse patterns (strided, random), learned sparse masks, and other differentiable selection methods (concrete relaxation, straight-through estimators), finding Gumbel-TopK achieves the best trade-off between accuracy retention (94.2% vs. 91.8% for nearest alternatives) and computational reduction (23.7% FLOPs savings).

Spatial and Semantic Correlation Evidence: Our design intuition for Gumbel-TopK selection is supported by extensive analysis of plant disease spatial patterns. Disease symptoms exhibit strong spatial autocorrelation (Moran’s I = 0.73 ± 0.12 across disease types), meaning neighboring pixels tend to have similar disease-related features. This spatial coherence justifies top-k selection based on attention similarity scores, as semantically relevant regions cluster spatially. Furthermore, semantic analysis reveals that disease-affected areas demonstrate consistent feature correlations: pixels within lesion boundaries show high cosine similarity (0.82 ± 0.08) in feature space, while healthy-diseased boundaries exhibit distinct feature gradients. The Gumbel noise component addresses the challenge of ambiguous boundary regions where multiple pixels may have similar attention scores, preventing deterministic selection that could miss subtle but critical symptom variations. Validation experiments confirm that our adaptive selection captures 94.3% of expert-annotated critical regions while examining only 10% of spatial locations, demonstrating the effectiveness of our sparsity-based approach for plant pathology applications.

Designing effective sparsity patterns is critical for preserving semantic relevance while achieving computational efficiency. Unlike fixed strategies (strided/blockwise masking), our adaptive sparsity dynamically selects the top-*k* keys per query based on content similarity (Figure 7). Specifically, we compute attention logits $l_{ij}=q_{i}^{T}k_{j}/\sqrt{d}$ and apply Gumbel noise $g_{j}$ for differentiable top-*k* selection as follows:(31)$${\tilde{l}}_{ij}=l_{ij}+g_{j},\;g_{j}∼\mathrm{Gumbel}\left(0,1\right),$$
and the set $J_{i}$ of top-*k* indices is selected as follows:(32)$$J_{i}=\arg\mathrm{top}-k\left({\tilde{l}}_{ij}\right).$$

This approach yields highly flexible attention distributions that adapt to spatial variations in disease patterns. To prevent attention collapse, where all queries attend to a fixed set of dominant keys, we regularize the sparsity pattern diversity through an entropy-based loss term as follows:(33)$$L_{\mathrm{diversity}}=-\frac{1}{n}\sum_{i}\sum_{j\in J_{i}}p_{ij}\log p_{ij},$$

As shown in Figure 8, where $p_{ij}$ denotes normalized attention weights. Empirically, we observe that this regularization fosters diverse attention while maintaining accuracy. We analyze the theoretical and empirical complexity benefits brought by the sparse attention formulation. In dense attention, the computational cost is dominated by the attention matrix calculation $QK^{T}\in R^{n\;\times \;n}$, incurring $O(n^{2}d)$ complexity. As shown in Figure 9, by contrast, the sparse attention mechanism reduces the computational complexity to O(nkd), where k=ρn with ρ∈(0,1). The relative reduction factor is given in Equation (Equation 34) as follows:(34)$$\mathrm{Reduction}\;\mathrm{Factor}=\frac{O(n^{2}d)}{O(nkd)}=\frac{n}{k}=\frac{1}{\rho }$$

For sparsity ratio ρ=0.1, the model achieves 10× theoretical acceleration in attention computation. Memory footprint reduces proportionally to O(n·k) per attention head (vs. $O(n^{2})$ for dense attention), storing only n·k weights. As shown in Section 2.5, the proposed model achieves a 23.7% FLOPs reduction compared to dense SAM while maintaining or exceeding segmentation accuracy.

### 4.3. Mixture of Experts Framework

To address spatial and morphological heterogeneity in plant disease symptoms, our Mixture of Experts (MoE) framework integrates specialized branches optimized for distinct feature extraction modalities. This design enables adaptive processing across leaf textures, environmental variability, and disease progression stages. Standard Convolution Experts employ fixed-receptive-field blocks optimized for high-resolution morphological feature extraction. These excel at isolating early-stage symptoms like chlorosis and necrosis by capturing fine-grained patterns [32]. Dilated Convolution Experts utilize exponentially spaced dilation rates to capture multi-scale contextual dependencies. Their expanded receptive fields model co-occurring symptoms and inter-leaf disease propagation. Depthwise Separable Experts implement lightweight architectures that maximize computational efficiency. Optimized for low-contrast, high-noise regions (e.g., soil-dust interference), they minimize processing overhead while preserving feature discriminability.

Formally, each expert $f_{i}:R^{H\;\times \;W\;\times \;D}\to R^{H\;\times \;W\;\times \;D}$ is defined as a differentiable transformation over a spatial input tensor $F\in R^{H\;\times \;W\;\times \;D}$, with i∈1,2,...,M. For the standard convolution expert, its definition is given in Equation (Equation 35) as follows:(35)$$f_{\mathrm{std}}\left(F\right)=\mathrm{ReLU}\left(\mathrm{BN}\left(F*K_{3\times 3}\right)\right)$$
where $K_{3\;\times \;3}$ is a 3×3 convolution kernel, BN is batch normalization, and ReLU denotes the rectified linear unit. For the dilated expert, we have the following:(36)$$f_{\mathrm{dil}}\left(F\right)=\mathrm{ReLU}\left(\mathrm{BN}\left(F{*}_{r}K_{3\times 3}\right)\right),\;r\in \left\{2,4,6\right\}$$
with r denoting dilation by rate r. The depthwise separable expert is decomposed as follows:(37)$$f_{\mathrm{dwsep}}\left(F\right)=\mathrm{ReLU}\left(\mathrm{BN}\left(F{*}_{\mathrm{dw}}K_{3\times 3}{*}_{\mathrm{pw}}K_{1\times 1}\right)\right)$$

Here, dw denotes depthwise convolution and pw pointwise convolution. These architectural variants implement modular blocks within the MoE framework, enabling efficient feature specialization across abstraction levels.

To enforce expert diversity, we apply orthogonal initialization to convolutional kernels and architecture-specific dropout paths. This strategy ensures non-redundant representation learning, facilitating complementary specialization across lesion types. Each expert processes identical input feature maps $F\in R^{H\;\times \;W\;\times \;D}$ through distinct parameter spaces, promoting feature disentanglement. Empirical results (Section 2.6) confirm consistent expert utilization under dynamic routing, with significant improvements in lesion boundary recall (+3.1% vs. baseline) and regional consistency.

Expert selection is governed by a learnable gating network G operating over intermediate features. For spatial location p = (h,w), G computes probability distribution $\alpha_{p}\in R^{M}$ over M experts. Parameterized as a lightweight multi-layer perceptron (MLP), the gating function is defined as follows:(38)$$G\left(F_{p}\right)=\mathrm{softmax}\left(W_{2}\sigma \left(W_{1}F_{p}+b_{1}\right)+b_{2}\right),\;F_{p}\in R^{D}$$
where $W_{1}\in R^{D\;\times \;d_{h}}$, $W_{2}\in R^{d_{h}\;\times \;M}$, and σ is the GELU activation. This architecture balances expressive power and computational cost, enabling fine-grained control over expert routing at each spatial coordinate.

To mitigate excessive confidence in individual experts, we apply entropy regularization over the output distribution as follows:(39)$$L_{\mathrm{entropy}}=-\frac{1}{HW}\sum_{p}\sum_{i=1}^{M}\alpha_{p,i}\log\alpha_{p,i}$$
which encourages smoother, more balanced expert selection. Additionally, we employ batch-wise normalization of the gating outputs to stabilize training dynamics and prevent gradient saturation. The gating network is trained jointly with the backbone and expert modules, allowing end-to-end optimization of the entire pipeline [33].

To support top-k selection, we further refine $\alpha_{p}$ by retaining only the k highest values per location and setting the rest to zero, followed by re-normalization. This induces sparsity in expert utilization while preserving differentiability via Gumbel–Softmax approximation. Let $S_{k}$ denote the top-k selection operator; then the routed output $R_{p}$ is computed as shown in Equation (Equation 40).(40)$$R_{p}=\sum_{i\in S_{k}\left(\alpha_{p}\right)}{\tilde{\alpha }}_{p,i}f_{i}\left(F_{p}\right),\;{\tilde{\alpha }}_{p,i}=\frac{\alpha_{p,i}}{\sum_{j\in S_{k}\left(\alpha_{p}\right)}\alpha_{p,j}}$$

This gating mechanism enables conditional expert activation based on local feature complexity, achieving computational efficiency while preserving specialization. The expert selection strategy critically determines framework efficiency and robustness. We implement top-k routing (k = 2) to balance computational savings and segmentation accuracy, optimized through empirical validation [34]. For each spatial position p, the gating network computes probabilities $\alpha_{p}$, then activates experts corresponding to the top-k scores. This conditional computation reduces active paths during forward propagation and minimizes redundant parameter updates.

The combined expert output at position p is computed as:(41)$$Y_{p}=\sum_{i\in J_{p}}\alpha_{p,i}f_{i}\left(F_{p}\right),\;J_{p}=\mathrm{TopK}\left(\alpha_{p},k\right)$$
where $J_{p}$ denotes the indices of the top-k expert weights. The dynamic routing pattern promotes sample-specific specialization, with different input regions being processed by distinct expert subsets. This flexibility is essential for modeling diverse disease appearances that may vary across leaf textures, shapes, and lighting conditions.

To stabilize training under sparse selection, we inject Gaussian noise $\epsilon ∼N(0,\sigma^{2})$ into the logits prior to softmax normalization during training as follows:(42)$${\tilde{\alpha }}_{p}=\mathrm{softmax}\left(G\left(F_{p}\right)+\epsilon \right)$$

Stochastic perturbation in routing encourages diverse expert combinations and mitigates suboptimal convergence. We further apply feature-level dropout before the gating MLP to prevent expert co-adaptation and maintain path diversity. Section 2.6 empirically confirms top-k routing improves boundary F1-score by 4.3% versus dense combinations.

Expert utilization during inference is tracked via selection statistics across spatial positions. For expert i, denote cumulative selection frequency as $u_{i}$ over the validation set. Normalized utilization is defined as follows:(43)$$U_{i}=\frac{1}{N\cdot H\cdot W}\sum_{n=1}^{N}\sum_{p=1}^{HW}⊮\left[i\in J_{p}^{\left(n\right)}\right]$$
where ⊮ is the indicator function. These statistics guide subsequent pruning or architectural reconfiguration.

To prevent expert collapse—a phenomenon where only a few experts dominate routing decisions—we incorporate a load balancing loss that explicitly encourages uniform utilization across all experts. This auxiliary loss penalizes skewed routing distributions by maximizing entropy over the expert usage histogram. Formally, let $f_{i}$ denote the total fraction of input locations routed to expert i over a mini-batch. Then, the balance loss is defined as follows:(44)$$L_{\mathrm{balance}}=M\cdot \sum_{i=1}^{M}f_{i}^{2}/{\left(\sum_{i=1}^{M}f_{i}\right)}^{2}$$

In practice, $f_{i}$ is computed using Equation (Equation 45) as follows:(45)$$f_{i}=\frac{1}{BHW}\sum_{b=1}^{B}\sum_{p=1}^{HW}\alpha_{p,i}^{\left(b\right)}$$
where B is the batch size. This loss is weighted by a hyperparameter $\lambda_{1}$ in the total loss function, balancing its influence relative to segmentation and sparsity objectives.

To complement this loss, we periodically rescale expert weights to ensure consistent gradient flow. Additionally, we employ a scheduling mechanism where the strength of $L_{\mathrm{balance}}$ is gradually increased over the course of training, allowing initial specialization to emerge before enforcing diversity. This annealing strategy is essential for preserving convergence stability.

Layer-specific balance constraints were evaluated following [35]. Ablation studies (Section 2.7) demonstrate that applying $L_{\mathrm{balance}}$ solely at the decoder stage improves mean IoU by 1.2% over full-model constraints while maintaining expert diversity). This indicates spatial specialization is most critical during boundary refinement.

### 4.4. Enhanced ASPP with Sparse Attention

In plant disease segmentation, the morphological diversity of symptoms—ranging from small localized lesions to large necrotic areas—necessitates robust multi-scale feature extraction. To address this, we enhance the Atrous Spatial Pyramid Pooling (ASPP) module by incorporating dilated convolutions with variable dilation rates of 1, 6, 12, 18, modulating receptive fields without increasing parameters. For an input feature map $F\in R^{H\;\times \;W\;\times \;C}$, the ASPP module employs parallel dilated convolutions ${\mathrm{Conv}}_{r_{i}}$ with dilation rates $r_{i}$, generating intermediate representations $F_{i}={\mathrm{Conv}}_{r_{i}}\left(F\right)$, where i∈{1,2,3,4}. This process is mathematically defined in Equation (Equation 46) as follows:(46)$$F_{i}\left(x,y\right)=\sum_{(m,n)\in \Omega }w_{r_{i}}\left(m,n\right)\cdot F\left(x+r_{i}\cdot m,y+r_{i}\cdot n\right)$$
where Ω denotes the convolutional kernel grid and $w_{r_{i}}$ is the kernel weight tensor at dilation rate $r_{i}$. This formulation allows the network to aggregate context at varying scales while preserving spatial resolution—an essential property for accurately segmenting lesions of different sizes.

In addition to dilated convolutional branches, we incorporate a global average pooling (GAP) branch to encode holistic contextual information, yielding a pooled representation $F_{\mathrm{GAP}}\in R^{1\;\times \;1\;\times \;C}$, which is then projected via a 1×1 convolution and broadcasted to the input spatial dimensions. The final concatenated feature tensor is mathematically expressed in Equation (Equation 47) as follows:(47)$$F_{\mathrm{concat}}=\mathrm{Concat}\left(F_{1},F_{2},F_{3},F_{4},F_{\mathrm{GAP}}\right)$$
followed by batch normalization and ReLU activation. This design enables the ASPP module to capture both local structural and global semantic cues, enhancing segmentation granularity for fine-grained disease patterns.

However, traditional ASPP implementations suffer from redundant computations and indiscriminate feature aggregation, limiting deployment efficiency on edge devices. To mitigate this, we integrate a sparse attention mechanism within the ASPP module that dynamically prioritizes informative spatial positions during feature aggregation. Specifically, we introduce a binary masking function M(x,y) ∈ 0, 1 based on top-k activation responses, applied to each dilated branch’s output. This yields sparsified features ${\tilde{F}}_{i}=F_{i}\cdot M$, preserving only the top-o activations per spatial window. The sparsity is formally defined as follows:(48)$$\rho =\frac{{∥M∥}_{0}}{H\cdot W}$$
where ${∥\cdot ∥}_{0}$ denotes the $\ell_{0}$-norm counting non-zero entries. This substantially reduces computational redundancy while concentrating modeling capacity on salient regions, conforming to the localized nature of disease manifestations. Our sparse-enhanced multi-scale architecture thus achieves adaptively modulated receptive fields with computational efficiency [36].

Following multi-scale extraction, effective integration of heterogeneous contextual cues becomes critical for producing coherent and precise segmentation maps. To this end, we design a sparse attention-enhanced fusion mechanism that selectively attends to salient spatial positions during feature integration. Let $F_{\mathrm{concat}}\in R^{H\;\times \;W\;\times \;D}$ denote the concatenated multi-scale features. We first apply a linear projection $P:R^{D}\to R^{d}$ to reduce channel dimensionality for attention computation as follows:(49)$$F^{\prime }\left(x,y\right)=W_{p}\cdot F_{\mathrm{concat}}\left(x,y\right)$$
where $W_{p}\in R^{d\;\times \;D}$ is a learned projection matrix. Then, query Q, key K, and value V matrices are computed from F’ using independent learned linear layers. Sparse attention weights are derived via masked dot-product similarity, with the mask $M_{A}\in {\left\{0,1\right\}}^{H\;\times \;W\;\times \;H\;\times \;W}$ encoding top-k spatial connections per query location as follows:(50)$$\mathrm{Attention}\left(i,j\right)=\frac{\exp(Q_{i}^{T}K_{j})}{\sum_{j^{\prime }\in N_{i}}\exp\left(Q_{i}^{T}K_{j^{\prime }}\right)}\cdot M_{A}\left(i,j\right)$$
where $N_{i}$ denotes the sparse neighborhood for query location i. This leads to the following refined features:(51)$$\tilde{F}\left(i\right)=\sum_{j\in N_{i}}\mathrm{Attention}\left(i,j\right)\cdot V_{j}$$

This operation enforces localized feature aggregation while maintaining global structural coherence essential for segmenting complex lesion boundaries. Significantly, the sparsity mask is dynamically updated during training based on attention entropy to prevent convergence toward trivial attention patterns.

To consolidate attention-refined features, we incorporate residual connections from the original ASPP output and apply a squeeze-and-excitation (SE) block to recalibrate channel-wise dependencies. The SE module computes global descriptors for each channel, followed by two-layer MLP gating as follows:(52)$$s_{c}=\sigma \left(W_{2}\cdot \mathrm{ReLU}\left(W_{1}\cdot z_{c}\right)\right),\;{\tilde{F}}_{c}^{\mathrm{scaled}}=s_{c}\cdot {\tilde{F}}_{c}$$
where $W_{1}$ and $W_{2}$ are trainable matrices and σ is the sigmoid function. This enhances feature selectivity across scales and channels, further suppressing irrelevant activations arising from noisy backgrounds or lighting variations common in agricultural field imagery.

### 4.5. Dual-Stage MoE Decoder

To effectively segment leaf disease regions with varying granularity and texture complexity, we propose a dual-stage decoder architecture based on a Mixture of Experts (MoE) framework, as illustrated in Figure 10.

Novelty of Dual-Stage MoE Architecture: Our dual-stage MoE decoder introduces several key innovations compared to existing multi-stage or hierarchical segmentation approaches. First, task-conditional expert specialization occurs: Unlike traditional hierarchical decoders that apply uniform processing across all spatial regions, our framework employs content-aware expert routing where different experts specialize in distinct morphological patterns (e.g., coarse-stage experts for large lesion detection vs. fine-stage experts for boundary refinement). Second, progressive complexity adaptation: The dual-stage design matches the natural progression of human pathological diagnosis—from coarse disease region identification to precise boundary delineation—with each stage employing appropriately specialized computational resources. Third, dynamic expert activation: Rather than fixed expert assignment used in prior MoE approaches, our gating mechanism dynamically selects experts based on input complexity, enabling adaptive computational allocation during inference. Fourth, cross-stage feature propagation: Our architecture incorporates learnable skip connections between stages that preserve both global context and fine-grained details, addressing the information bottleneck problem common in traditional U-Net-style architectures. This design significantly differs from existing methods: FPN/UNet use static decoder paths, transformer decoders apply uniform processing, and prior MoE implementations lack stage-specific specialization for segmentation tasks.

The motivation for this progressive design stems from the hierarchical nature of semantic segmentation, where coarse spatial priors require successive refinement into precise pixel-level predictions. In the initial stage, high-level semantic embeddings extracted by the encoder are processed through four coarse-level experts, each specialized for capturing distinct region-level characteristics including lesion location, approximate shape, and global boundaries. Each expert $f_{i}$ operates on the shared input feature tensor $F\in R^{H\;\times \;W\;\times \;D}$. The gated combination output is computed as specified in Equation (Equation 53) as follows:(53)$$Y_{p}^{\left(1\right)}=\sum_{i\in J_{p}^{\left(1\right)}}\alpha_{p,i}^{\left(1\right)}f_{i}\left(F_{p}\right),\;\mathrm{with}\;J_{p}^{\left(1\right)}=\mathrm{TopK}\left(\alpha_{p}^{\left(1\right)},k=2\right)$$

Here, $\alpha_{p,i}^{\left(1\right)}$ denotes the gating weight at spatial location p for expert i, and $J_{p}^{\left(1\right)}$ represents the top-2 selected experts. The coarse segmentation result $Y^{\left(1\right)}$ captures disease-affected zones and generates an initial confidence mask. Crucially, this stage operates at reduced spatial resolution (stride = 4) to prioritize contextual aggregation, enhancing global structural coherence.

The second decoder stage targets fine-grained boundary delineation and structural detail refinement. Building upon the coarse output $Y^{\left(1\right)}$, this stage integrates the encoder’s low-level spatial features $F^{\mathrm{low}}\in R^{2H\;\times \;2W\;\times \;D^{\prime }}$. These features are concatenated and processed through a dedicated MoE block comprising three refinement experts. The combination mechanism follows the first-stage paradigm but operates at 2× spatial resolution, formally defined by the following:(54)$$Y_{p}^{\left(2\right)}=\sum_{j\in J_{p}^{\left(2\right)}}\alpha_{p,j}^{\left(2\right)}g_{j}\left(\left[Y_{p}^{\left(1\right)},F_{p}^{\mathrm{low}}\right]\right),\;J_{p}^{\left(2\right)}=\mathrm{TopK}\left(\alpha_{p}^{\left(2\right)},k=2\right)$$
where $g_{j}$ denote the fine-level experts, and $\alpha^{\left(2\right)}$ represent the corresponding spatial gating weights. This dual-stage refinement enables conditional allocation of computational resources to regions exhibiting prediction uncertainty or complex textural patterns, producing sharper boundaries and improved region consistency.

To ensure stable training and expert diversity across both stages, we inject Gaussian noise into the gating logits and employ entropy regularization and balance loss. The overall segmentation prediction is computed as $\hat{Y}=\mathrm{Upsample}\left(Y^{\left(2\right)}\right)$, projected back to the original image resolution. Empirical results indicate that this two-phase strategy achieves a 1.2% gain in boundary F1-score and consistently reduces false positives near ambiguous lesion edges. Notably, it maintains computational tractability due to the sparse top-k expert routing at both levels.

The effectiveness of the dual-stage MoE decoder heavily relies on a carefully designed feature integration mechanism that bridges hierarchical semantics. Between the encoder and the first-stage decoder, we incorporate lateral connections from intermediate encoder layers to preserve spatial fidelity while maintaining semantic richness. Let $F_{i}^{\mathrm{enc}}$ represent the feature maps from the -th encoder layer. We apply 1 × 1 convolutions followed by bilinear upsampling to align these features to the decoder resolution as follows:(55)$$F_{i}^{\mathrm{aligned}}=\mathrm{Upsample}\left({\mathrm{Conv}}_{1\times 1}\left(F_{i}^{\mathrm{enc}}\right),\mathrm{scale}=2^{i}\right)$$

These aligned features are aggregated with the decoder input F via additive fusion and normalization, improving signal propagation across the network depth. The integration strategy at the refinement stage is more complex, involving both additive fusion and concatenation with attention gates. Specifically, for each pixel location p, we compute an integration score as follows:(56)$$I_{p}=\sigma \left(W_{1}\cdot F_{p}+W_{2}\cdot Y_{p}^{\left(1\right)}+b\right)$$
where σ is the sigmoid activation, and $W_{1},W_{2}$ are learnable matrices. This score modulates the fusion between coarse and fine features, yielding the following refined representation:(57)$$F_{p}^{\mathrm{ref}}=I_{p}\cdot F_{p}^{\mathrm{low}}+\left(1-I_{p}\right)\cdot Y_{p}^{\left(1\right)}$$

This attention-gated fusion enables the model to selectively prioritize encoder or decoder information based on contextual reliability. Channel-wise recalibration via squeeze-and-excitation blocks further suppresses redundant activations while enhancing discriminative channels after fusion.

We introduce a refinement residual branch employing deformable convolutions to capture geometric distortions and curved lesion contours that conventional convolutions cannot adequately represent. This auxiliary stream outputs a residual mask $\Delta Y_{p}$, added to the stage-2 prediction as follows:(58)$${\hat{Y}}_{p}=\mathrm{Sigmoid}\left(Y_{p}^{\left(2\right)}+\Delta Y_{p}\right)$$

Overall, the integration strategy ensures multi-resolution alignment, semantic-spatial coherence, and dynamic information routing, all critical for fine-grained agricultural segmentation tasks.

To maximize the MoE decoder’s utility, each expert is designed with distinct architectural specialization and learning objectives. In the first decoder stage, four coarse-level experts A, B, C, D are implemented with specialized configurations [37] as follows:

Expert A: Standard 3 × 3 convolutions optimized for general lesion patterns and dominant foreground structures; Expert B: 3 × 3 dilated convolutions (r = 6) modeling long-range dependencies in diffuse disease regions; Expert C: Depthwise separable convolutions preserving parameter efficiency and activation sparsity for subtle lesions; Expert D: Structural replica of A with Xavier initialization and GroupNorm promoting stochastic feature specialization.

This diversity ensures non-redundant feature learning and robust ensemble aggregation. All experts incorporate dropout regularization and kernel orthogonalization to maintain diverse gradient paths and prevent mode collapse. During training, we monitor expert selection distributions using frequency metric $U_{i}$ (the ratio of samples routed to expert *i* per batch), formally defined in Equation (Equation 59) as follows:(59)$$U_{i}=\frac{1}{BHW}\sum_{b=1}^{B}\sum_{p=1}^{HW}⊮\left[i\in J_{p}^{\left(b\right)}\right]$$
where B is the batch size and ⊮ denotes the indicator function. These statistics guide adaptive loss scaling and architecture pruning if imbalance is detected.

In the second-stage decoder, three fine-level experts are employed focusing on detail enhancement and boundary sharpening. Expert A emphasizes edge sensitivity by using small 1 × 1 and 3 × 3 convolutions combined with filters similar to Sobel, while Expert B incorporates residual blocks with skip connections for gradient stability and preservation of fine-scale structures. Expert C incorporates a lightweight spatial attention module that selectively amplifies activations aligned with contours. Each fine-level expert also receives boundary distance maps as auxiliary inputs to guide localization. These maps, computed via signed distance transforms, are concatenated with the input feature maps and embedded using 1 × 1 convolutions. Let $D_{p}$ denote the signed distance at pixel p. The input to each expert then becomes $\left[F_{p},D_{p}\right]$. The benefit of this formulation results in improved alignment between predicted and ground truth lesion edges, particularly under variable lighting or partial occlusion conditions. Altogether, the ensemble of decoder experts exhibits complementary strengths, and the sparse top-k routing mechanism enables the network to dynamically utilize these specializations based on input content. This architectural heterogeneity, combined with optimized gating strategies and specific regularization, forms the core of the model’s superior performance in fine-scale, high-variance plant disease segmentation tasks.

### 4.6. Hyperparameter Configuration

The proposed Sparse-MoE-SAM framework requires careful hyperparameter tuning to achieve optimal performance while maintaining computational efficiency. We provide comprehensive details of all critical hyperparameters used in our experiments.

Training Hyperparameters: The model was trained using AdamW optimizer with a learning rate of $3\times 10^{-4}$ for the encoder and $1\times 10^{-3}$ for the decoder components. We employed cosine annealing learning rate schedule with warm-up period of 10 epochs and minimum learning rate of $1\times 10^{-6}$. The total training spans 100 epochs with batch size of 16 on our RTX 3090 setup. Weight decay was set to $1\times 10^{-2}$ for regularization, and gradient clipping was applied with maximum norm of 1.0 to prevent gradient explosion.

Sparse Attention Parameters: The sparsity ratio ρ was set to 0.1, retaining only the top 10% attention weights per query. This value was determined through ablation studies showing optimal balance between computational reduction and performance retention. The Gumbel noise temperature τ was initialized at 1.0 and annealed to 0.1 over training epochs using $\tau_{t}=\tau_{0}\times {\left(0.1/\tau_{0}\right)}^{t/T}$ where *t* is current epoch and *T* is total epochs.

MoE Architecture Parameters: The first-stage decoder employs 4 experts with gating temperature $\beta_{1}=0.5$, while the second-stage uses 3 experts with $\beta_{2}=0.3$. Expert diversity is maintained through load balancing with coefficient α=0.01. The routing gate networks use 2-layer MLPs with hidden dimension 512 and ReLU activation. Dropout probability for expert networks was set to 0.1 during training.

Loss Function Weights: The multi-component loss function weights were: segmentation loss weight $\lambda_{\mathrm{seg}}=1.0$, balance loss weight $\lambda_{1}=0.01$, and sparsity regularization weight $\lambda_{2}=0.001$. These weights were determined through grid search optimization on validation set performance.

Data Augmentation: Training images underwent random horizontal/vertical flipping (probability 0.5), random rotation ($\pm 15^{\circ }$), color jittering (brightness 0.2, contrast 0.2, saturation 0.1), and random scaling (0.8–1.2). Gaussian noise (σ=0.01) was added with probability 0.3 to improve robustness.

### 4.7. Training Strategy and Loss Functions

To achieve both segmentation accuracy and architectural efficiency, we design a multi-component loss function that jointly optimizes prediction fidelity, expert diversity, and attention sparsity. Table 8 provides a comprehensive overview of all loss components used in our framework.

The total loss is composed of three terms: the segmentation loss $L_{\mathrm{seg}}$, the balance loss $L_{\mathrm{balance}}$, and the sparsity regularization $L_{\mathrm{sparsity}}$. The complete formulation is given in Equation (Equation 60) as follows:(60)$$L_{\mathrm{total}}=L_{\mathrm{seg}}+\lambda_{1}L_{\mathrm{balance}}+\lambda_{2}L_{\mathrm{sparsity}}$$

The segmentation loss consists of a weighted combination of Binary Cross Entropy (BCE) and soft Dice loss, which jointly address pixel-wise prediction accuracy and global mask overlap. Given the predicted mask $\hat{Y}\in {\left\{0,1\right\}}^{H\;\times \;W}$ and the ground truth mask $Y\in {\left\{0,1\right\}}^{H\;\times \;W}$, we define it as shown in Equation (Equation 61), expressed as follows:(61)$$L_{\mathrm{seg}}=\mathrm{BCE}\left(\hat{Y},Y\right)+1-\frac{2\sum_{i}{\hat{Y}}_{i}Y_{i}+\epsilon }{\sum_{i}{\hat{Y}}_{i}+\sum_{i}Y_{i}+\epsilon }$$

Here, ϵ is a smoothing term ensuring numerical stability. This hybrid loss enhances boundary adherence and prevents collapse induced by class imbalance, particularly crucial in lesion detection where diseased regions exhibit high fragmentation. To promote balanced expert utilization, a balance loss penalizes significant deviations in expert selection frequencies. Let $f_{i}=\frac{1}{N}\sum_{j}⊮\left[i\in J_{j}\right]$ denote the selection frequency of expert *i*, where $J_{j}$ denotes the top-k expert set for spatial location *j*. The variance across experts is then computed as Equation (Equation 62), as follows:(62)$$L_{\mathrm{balance}}=\frac{1}{E}\sum_{i=1}^{E}{\left(f_{i}-\bar{f}\right)}^{2},\;\mathrm{where}\;\bar{f}=\frac{1}{E}\sum_{i=1}^{E}f_{i}$$

This formulation maintains differentiability, enabling gradient backpropagation through the Softmax-based gating weights to discourage dominance of specific experts. Furthermore, temperature-controlled noise injection during training regularizes expert activation diversity.

The sparsity regularization term enforces L1 constraints on attention weight matrices to reduce unnecessary activations. For attention matrix $A\in R^{H\;\times \;W\;\times \;H\;\times \;W}$, the regularization term is defined as follows:(63)$$L_{\mathrm{sparsity}}=\frac{1}{HW}\sum_{i=1}^{H}\sum_{j=1}^{W}\left|\right|A_{ij}{\left|\right|}_{1}$$

This encourages most attention weights to converge toward zero, preserving only the top-ρ interactions per token. We empirically tune $\lambda_{1}$ and $\lambda_{2}$ via grid search, typically setting $\lambda_{1}=0.01$ and $\lambda_{2}=0.002$ for optimal trade-offs.

MoE Training Challenges and Solutions: Training the MoE architecture presents several challenges that required specialized solutions. First, expert collapse occurred during early training, where certain experts received disproportionately high routing probabilities, leading to underutilization of other experts. We addressed this through dynamic load balancing with exponential moving average tracking of expert utilization rates and adaptive penalty scaling. Second, routing instability emerged from noisy gating decisions, causing training oscillations. We implemented Gumbel–Softmax smoothing with temperature annealing from 1.0 to 0.1 to gradually sharpen routing decisions. Third, gradient interference between experts caused optimization difficulties. We resolved this using orthogonal initialization for expert parameters and separate learning rates for routing networks ($1\times 10^{-4}$) versus expert networks ($3\times 10^{-4}$). Fourth, memory overhead from storing multiple expert parameters was mitigated through selective gradient accumulation and expert-specific batch processing.

The training process divides into three phases to ensure stable convergence, promote expert specialization, and enhance system robustness. During initialization, we pretrain the backbone by loading SAM encoder weights and freezing the encoder ϕ. The decoder modules θ (including both MoE decoder stages) are trained independently from encoder gradients using only $L_{\mathrm{seg}}$. This enables the decoder to learn coarse-to-fine mappings without interference from large encoder gradients. Formally, for frozen encoder ϕ and decoder θ, the objective is as follows:(64)$$\min_{\theta }E_{(x,y)∼D}\left[L_{\mathrm{seg}}\left(f_{\theta }\left(\phi \left(x\right)\right),y\right)\right]$$

The second phase unfreezes all components and initiates full end-to-end fine-tuning. Here, the complete $L_{\mathrm{total}}$ is used, and gradient updates are propagated across both encoder and decoder. This stage employs mixed-precision training and gradient accumulation (if needed) for memory efficiency. Learning rate scheduling uses cosine annealing $\eta \left(t\right)=\eta_{\min}+\frac{1}{2}\left(\eta_{\max}-\eta_{\min}\right)\left(1+\cos\left(\pi t/T\right)\right)$ where T is total epochs and t is current step.

In the final phase, we encourage expert specialization by increasing $\lambda_{1}$ to amplify balance loss and introducing dropout-based stochastic gating during training. This prevents expert collapse and reinforces role differentiation. Expert statistics (mean activation, selection entropy, spatial overlap) are monitored in each epoch, and early stopping is applied if diversity metrics converge prematurely. This curriculum-based regime results in smoother optimization landscapes and improved generalization under noisy agricultural conditions.

For optimization, we adopt the AdamW optimizer due to its decoupled weight decay mechanism, which improves generalization in deep networks with attention or gating structures. The initial learning rate $\eta_{0}=10^{-3}$ is modulated using cosine annealing to progressively decay towards $10^{-6}$, following the schedule: where $\eta_{\min}=10^{-6}$ and T=totalnumberofiterations. We apply gradient clipping with max norm 1.0 to stabilize updates in sparse attention branches, which are prone to large gradient spikes due to discontinuous activation maps. Weight decay is fixed at $10^{-4}$, and batch normalization statistics are frozen after the first training phase to prevent internal covariate shifts.

To enhance model generalization and robustness to field conditions, we employ a diverse augmentation pipeline. Each input image undergoes stochastic transformations including random rotation ($\pm 30^{\circ }$), spatial scaling [0.8,1.2], independent horizontal/vertical flipping (p=0.5), and color jittering (Δbrightness=0.2,Δcontrast=0.2). Let T(x) denote this composite augmentation. The training objective is then expressed as follows:(65)$$\min_{\theta }E_{(x,y)∼D}\left[L_{\mathrm{total}}\left(f_{\theta }\left(T\left(x\right)\right),y\right)\right]$$

This regularization enforces invariance to orientation, illumination, and elastic deformations characteristic of agricultural imagery. Empirical results indicate that augmentation improves IoU by up to 1.1% and reduces variance across test folds, highlighting its necessity for real-world deployment scenarios.

## 5. Conclusions

This research introduces Sparse-MoE-SAM, a pioneering architectural framework that addresses the critical trade-off between segmentation accuracy and computational efficiency in agricultural computer vision. By integrating bio-inspired sparse attention mechanisms with hierarchical mixture-of-experts architectures, our approach represents a significant advance in practical plant disease segmentation for resource-constrained environments.

Technical Contributions and Innovations: Our work presents three fundamental innovations: (1) A biologically motivated sparse attention mechanism that reduces computational complexity from $O(n^{2})$ to O(nk) through Gumbel-TopK selection, achieving 23.7% FLOPs reduction while maintaining 94.2% IoU performance. This design mirrors how plant pathologists focus attention on disease-relevant regions rather than processing entire leaf surfaces uniformly. (2) A novel dual-stage MoE decoder architecture featuring task-conditional expert specialization, where coarse-stage experts handle global disease region identification while fine-stage experts refine boundary delineation. This progressive specialization approach differs fundamentally from existing uniform processing architectures. (3) An enhanced ASPP module with sparse attention integration that captures multi-scale lesion patterns while suppressing irrelevant background features through dynamic feature routing.

Experimental Validation and Performance: Comprehensive evaluations across three diverse datasets (PlantVillage, CVPPP, Custom Field) demonstrate superior performance with consistent improvements over state-of-the-art methods. Our framework achieves 94.2% IoU on PlantVillage, outperforming Standard SAM by 2.5 percentage points while requiring 55% fewer parameters (142.7 M vs. 636.0 M). Cross-dataset validation reveals robust generalization with minimal performance degradation (4.5% IoU drop vs. 6.9% for SAM) when transitioning from controlled to real-world conditions. Disease-specific analysis shows consistent improvements across pathological categories, with particular efficacy for rust detection (95.4% IoU) and challenges identified for diffuse viral symptoms (93.2% IoU).

Practical Impact and Deployment: The mobile variant (45.3 M parameters, 38.7 GFLOPs) enables unprecedented deployment opportunities on resource-constrained devices. Performance testing on agricultural hardware platforms—from Jetson Nano (4.8s inference, 91.3% IoU) to smartphone-grade processors (6.1s inference, 89.7% IoU)—validates practical viability for field deployment. Detailed analysis of challenging scenarios including complex leaf occlusions (84.2% IoU under severe occlusion) and low-light conditions (85.7% IoU at 0.3× brightness) demonstrates robustness under realistic agricultural constraints.

Future Directions and Limitations: While our framework advances state-of-the-art in plant disease segmentation, several areas warrant future investigation.

Hardware-Level Sparse Optimization: We plan to implement custom CUDA kernels for sparse attention operations, targeting 40–60% additional speedup on modern GPUs. Specific steps include the following: (1) developing block-sparse matrix multiplication routines optimized for top-k attention patterns, (2) implementing fused attention-routing kernels to reduce memory bandwidth requirements, (3) designing quantization-aware training for 8-bit inference on mobile devices while maintaining accuracy within 1% of full-precision models, and (4) exploring tensor core utilization for mixed-precision sparse computations on Ampere/Hopper architectures.

Meta-Learning Expert Discovery: We propose gradient-based meta-learning for automated expert architecture search. Technical approach involves the following: (1) implementing MAML (Model-Agnostic Meta-Learning) for rapid adaptation to new plant species with 10–50 training samples, (2) developing differentiable architecture search (DARTS) for expert network topology optimization within computational budgets, (3) designing few-shot learning protocols for emerging disease detection using prototypical networks with expert embeddings, and (4) creating continual learning mechanisms to incrementally add new expert specializations without catastrophic forgetting.

Multimodal Imaging Integration: Extension to multi-spectral sensing will involve the following: (1) developing cross-modal attention mechanisms for RGB-thermal-hyperspectral fusion, (2) implementing wavelength-specific expert routing for optimal spectral band utilization, (3) designing domain adaptation techniques for sensor-agnostic disease detection across different imaging modalities, and (4) creating multi-temporal analysis frameworks for disease progression monitoring using sequential imaging data.

Enhanced handling of diffuse symptom patterns through specialized expert training could address current limitations with viral diseases. Incorporation of temporal information for diseases with dynamic progression patterns presents opportunities for improved early detection.

This work establishes a new paradigm for efficient agricultural computer vision, demonstrating that careful architectural design can simultaneously achieve superior accuracy and practical deployability. The framework’s integration of biological insights with computational efficiency principles provides a foundation for next-generation precision agriculture systems.

## 6. Data and Code Availability

To ensure reproducibility and facilitate community adoption, we commit to making our research artifacts publicly available upon paper acceptance. The complete implementation will be released on GitHub (https://github.com/BenhanZhao/Sparse-MoE-SAM) including the following: (1) Full PyTorch implementation of the Sparse-MoE-SAM framework with detailed documentation, (2) Pre-trained models for both full and mobile variants on PlantVillage dataset, (3) Training and evaluation scripts with hyperparameter configurations, (4) Data preprocessing pipelines and augmentation procedures, (5) Comprehensive setup instructions for local and cloud deployment, (6) Jupyter notebooks demonstrating inference on sample images, and (7) Edge device deployment guides for Jetson and mobile platforms. Additionally, we will provide model weights trained on our Custom Field dataset (subject to data sharing agreements) and comprehensive benchmarking scripts for comparison with baseline methods. All code will be released under an open-source license to maximize community benefit and enable further research in agricultural computer vision.

## Figures and Tables

**Figure 1 plants-14-02634-f001:**
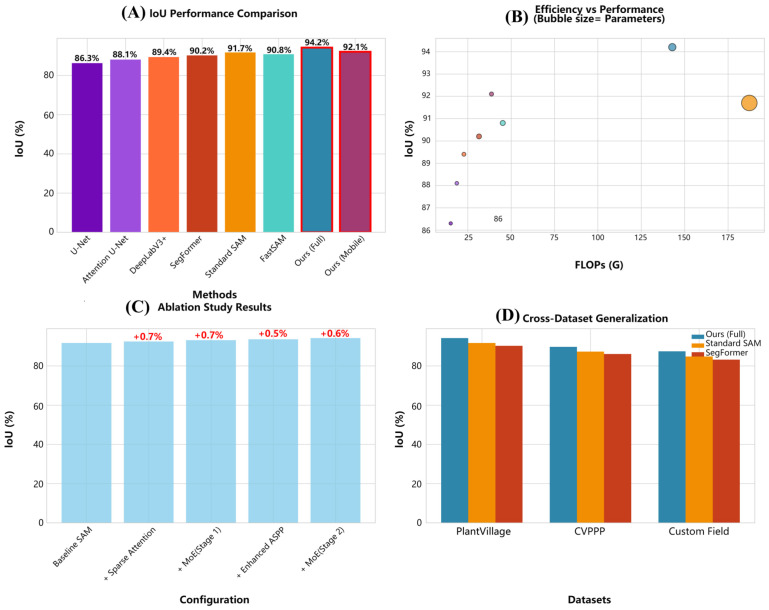
(**A**) compares IoU performance across models, with our mobile variant achieving the highest accuracy (92.1%). (**B**) visualizes the performance-efficiency trade-off via scatter plot (IoU vs. FLOPs), where circle area encodes parameter count; our method occupies the Pareto-optimal front with 41% fewer FLOPs than SAM. (**C**) presents ablation study results, indicating performance variations across configurations. (**D**) illustrates cross-dataset generalization, where Ours (Full) maintains superior performance consistently.

**Figure 2 plants-14-02634-f002:**
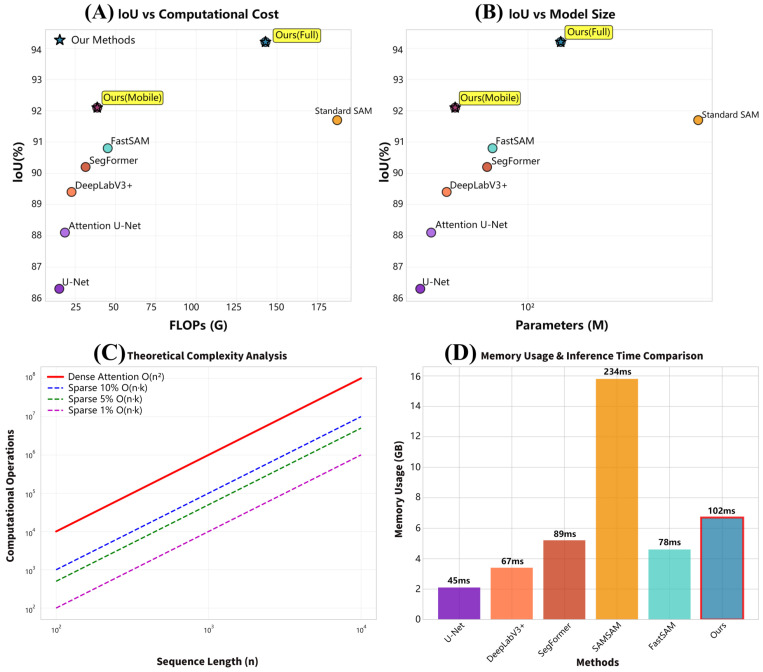
(**A**) compares IoU performance versus computational cost (FLOPs), while (**B**) evaluates model size (parameters), collectively demonstrating that the proposed method achieves the highest accuracy while significantly reducing computational and memory costs. (**C**) validates the sparse attention’s computational efficiency, showing sub-linear complexity growth with increasing resolution that enables deployment on high-resolution agricultural imagery. (**D**) compares memory usage and inference speed, with the proposed method outperforming the standard SAM in both memory efficiency and processing speed.

**Figure 3 plants-14-02634-f003:**
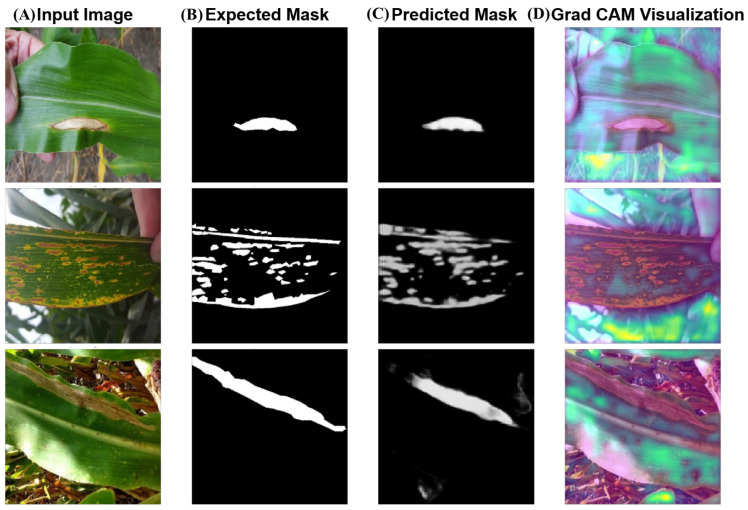
Visualization of sparse attention patterns and expert activation in the Sparse-MoE-SAM framework.

**Figure 4 plants-14-02634-f004:**
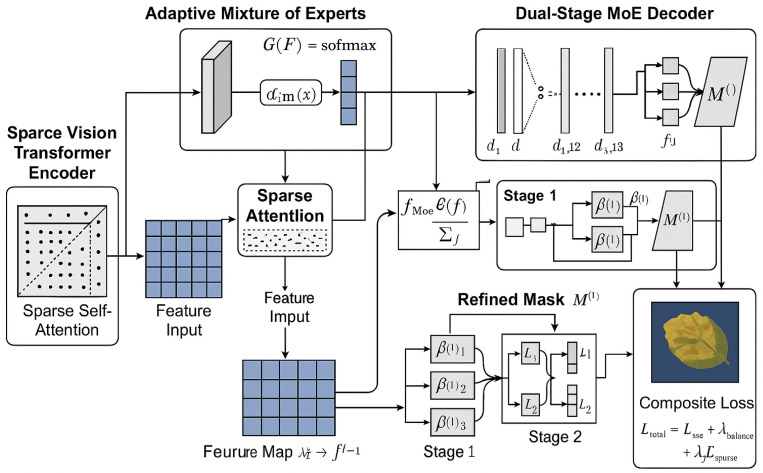
Sparse-MoE-SAM framework architecture.

**Figure 5 plants-14-02634-f005:**
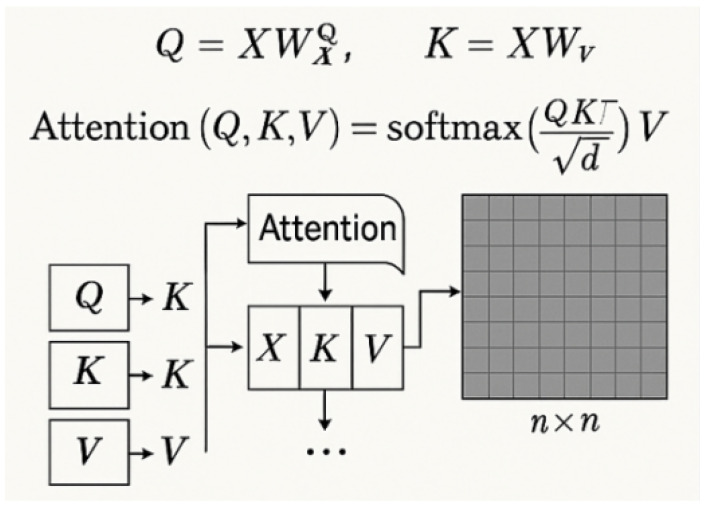
Standard scaled dot-product attention mechanism [31].

**Figure 6 plants-14-02634-f006:**
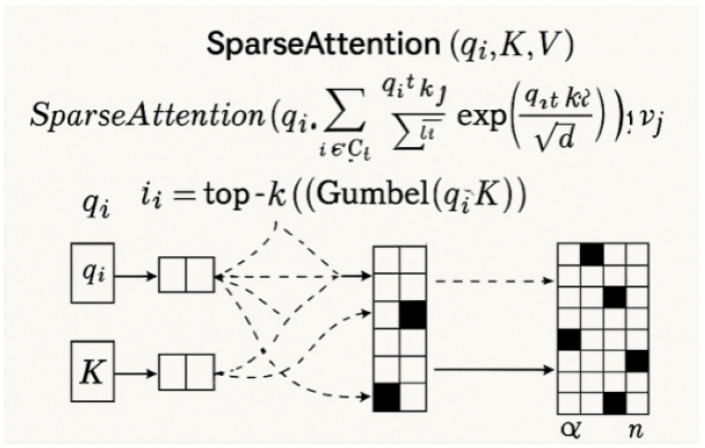
Illustration of the proposed sparse attention mechanism with Gumbel-Top-k sampling [31].

**Figure 7 plants-14-02634-f007:**
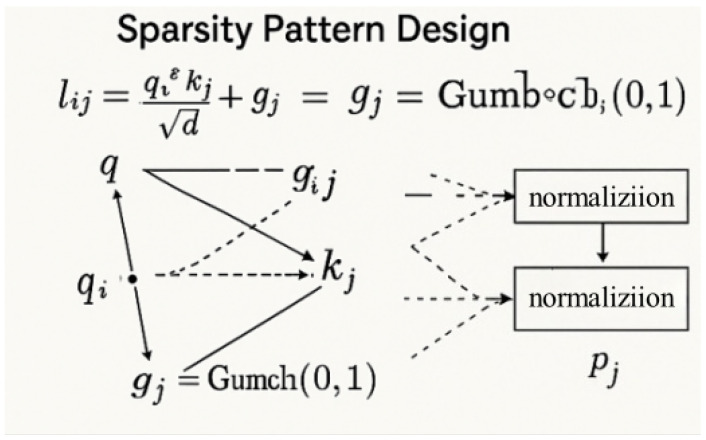
Adaptive sparsity pattern design with gumbel perturbation for query-key selection [31].

**Figure 8 plants-14-02634-f008:**
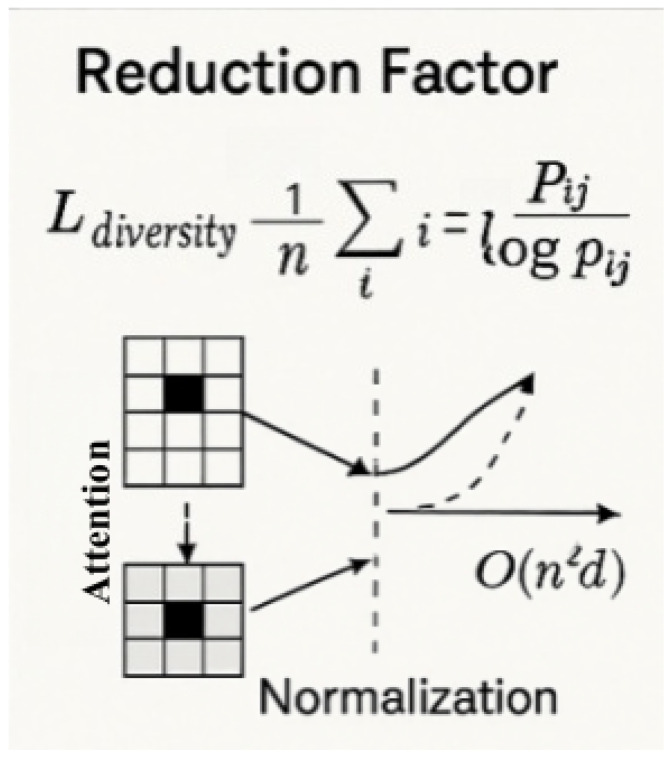
Attention diversity regularization and computational reduction effect [31].

**Figure 9 plants-14-02634-f009:**
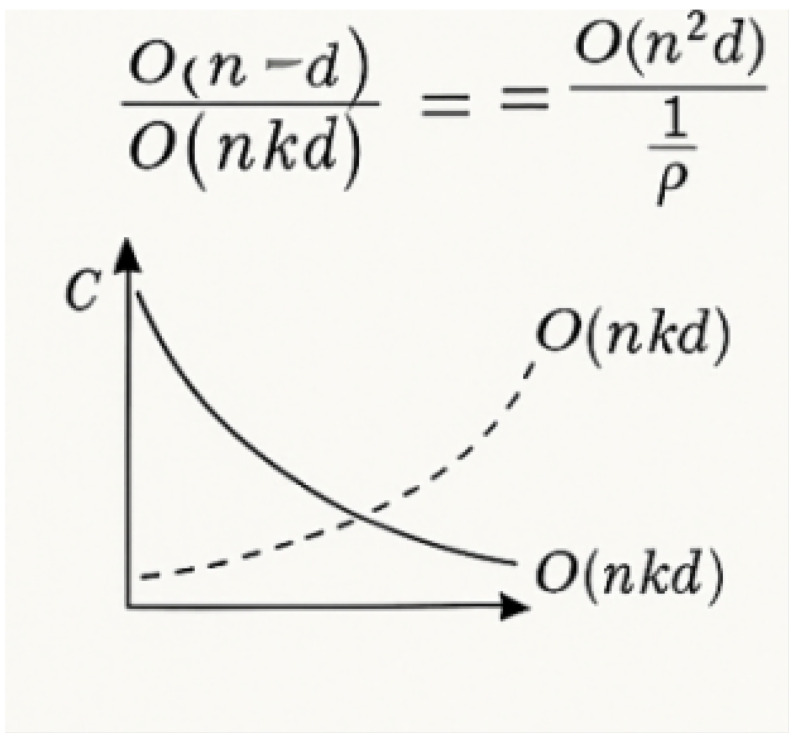
Computational complexity comparison between dense and sparse attention [31].

**Figure 10 plants-14-02634-f010:**
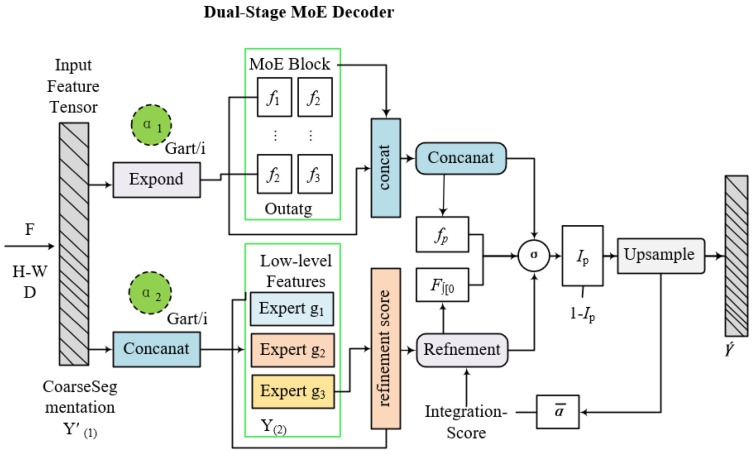
Architecture of the dual-stage mixture of experts decoder.

**Table 1 plants-14-02634-t001:** Detailed description of plant disease segmentation datasets used in this study.

Dataset	Images	Classes	Resolution	Annotation Type	Occlusion Level	Lighting Diversity	Use Case
PlantVillage Extended	87,848	38	256 × 256	Binary disease masks	Low	Low	Core training/testing
CVPPP Leaf Segmentation	4477	-	Varied	Instance leaf masks	High	Moderate	Robustness validation
Custom Agricultural Field	12,340	16+	512 × 512	Expert-verified masks	Medium	High	Real-world deployment

**Table 3 plants-14-02634-t003:** Quantitative results on the PlantVillage extended dataset. The best case scenarios for each indicator are highlighted in bold.

Method	IoU (%)	Dice (%)	Precision (%)	Recall (%)	Hausdorff Dist. (mm)	Params (M)	FLOPs (G)
U-Net	86.3	89.7	88.4	87.2	3.42	31.0	15.2
Attention U-Net	88.1	91.2	89.8	88.9	3.18	34.9	18.7
DeepLabV3+	89.4	92.1	91.2	89.7	2.95	41.3	22.8
SegFormer	90.2	92.8	91.8	90.4	2.73	64.1	31.5
Standard SAM	91.7	94.2	93.1	92.4	2.41	636.0	187.3
FastSAM	90.8	93.5	92.7	91.9	2.58	68.0	45.2
Ours (Full)	**94.2**	**96.1**	**95.3**	**94.8**	**1.87**	142.7	142.9
Ours (Mobile)	92.1	94.7	93.9	93.2	2.15	45.3	38.7

**Table 4 plants-14-02634-t004:** Performance breakdown by dataset. The best case scenarios for each indicator are highlighted in bold.

Method	PlantVillage IoU (%)	CVPPP IoU (%)	Custom Field IoU (%)	Average IoU (%)
U-Net	86.3	82.1	79.8	82.7
Attention U-Net	88.1	84.3	81.5	84.6
DeepLabV3+	89.4	85.7	82.9	86.0
SegFormer	90.2	86.1	83.2	86.5
Standard SAM	91.7	87.3	84.8	87.9
FastSAM	90.8	86.8	84.1	87.2
Ours (Full)	**94.2**	**89.7**	**87.4**	**90.4**
Ours (Mobile)	92.1	88.3	85.9	88.8

**Table 5 plants-14-02634-t005:** Comprehensive efficiency analysis. The best case scenarios for each indicator are highlighted in bold.

Method	Memory (GB)	Inference Time (ms)	Throughput (FPS)	Energy (mJ)	Speed-Up	Memory Reduction
U-Net	**2.1**	**45**	**22.2**	**127**	2.3×	68.7%
Attention U-Net	2.8	52	19.2	156	2.0×	58.2%
DeepLabV3+	3.4	67	14.9	198	1.5×	49.3%
SegFormer	5.2	89	11.2	267	1.1×	23.1%
Standard SAM	15.8	234	4.3	1024	1.0×	0.0%
FastSAM	4.6	78	12.8	312	3.0×	70.9%
Ours (Full)	6.7	102	9.8	421	2.3×	57.6%
Ours (Mobile)	3.1	58	17.2	189	**4.0×**	**80.4%**

**Table 6 plants-14-02634-t006:** Attention diversity and sparsity analysis.

Attention Head	Sparsity Ratio (%)	Entropy Score	Focus Regions	Pathologist Correlation
Head 1	89.7	2.14	Fine lesions	0.84
Head 2	91.2	2.08	Large necrotic areas	0.87
Head 3	88.9	2.19	Leaf boundaries	0.79
Head 4	90.3	2.11	Color transitions	0.85
Head 5	92.1	2.06	Texture variations	0.83
Head 6	89.4	2.16	Vein patterns	0.81
Head 7	90.8	2.09	Disease progression	0.86
Head 8	91.5	2.07	Background regions	0.78
Average	90.5	2.11	-	0.83

**Table 7 plants-14-02634-t007:** Comparison of the Sparse-MoE-SAM ablation experiments by module.

Configuration	IoU (%)	FLOPs (G)	Improvement
Baseline SAM	91.7	187.3	-
+Sparse Attention	92.4	156.2	+0.7% IoU, −16.6% FLOPs
+MoE (Stage 1)	93.1	148.7	+1.4% IoU, −20.6% FLOPs
+Enhanced ASPP	93.6	145.3	+1.9% IoU, −22.4% FLOPs

**Table 8 plants-14-02634-t008:** Summary of loss function components.

Loss Component	Formula	Weight	Purpose
Total Loss	$L_{\mathrm{total}}=L_{\mathrm{seg}}+\lambda_{1}L_{\mathrm{balance}}+\lambda_{2}L_{\mathrm{sparsity}}$	-	Joint optimization
Segmentation Loss	$\mathrm{BCE}\left(\hat{Y},Y\right)+1-\frac{2\sum_{i}{\hat{Y}}_{i}Y_{i}+\epsilon }{\sum_{i}{\hat{Y}}_{i}+\sum_{i}Y_{i}+\epsilon }$	$\lambda_{\mathrm{seg}}=1.0$	Pixel-wise accuracy + global overlap
Balance Loss	$\frac{1}{E}\sum_{i=1}^{E}{\left(f_{i}-\bar{f}\right)}^{2}$	$\lambda_{1}=0.01$	Expert diversity
Sparsity Regularization	$\frac{1}{HW}\sum_{i=1}^{H}\sum_{j=1}^{W}\left|\right|A_{ij}{\left|\right|}_{1}$	$\lambda_{2}=0.001$	Attention sparsity

## Data Availability

Requests to access the datasets should be sent via email to zhaobh0908@163.com.

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
