# Peer review of "Sparse-MoE-SAM: A Lightweight Framework Integrating MoE and SAM with a Sparse Attention Mechanism for Plant Disease Segmentation in Resource-Constrained Environments"

_plants, 2025, doi:10.3390/plants14172634_

Round 1

Reviewer 1 Report

Comments and Suggestions for Authors

In this study, a lightweight deep learning framework for plant disease segmentation that combines sparse attention and a Mixture of Experts (MoE) decoder to achieve high accuracy and low computational cost was proposed.

There are some similarities in different sections of the manuscript. Please either remove or paraphrase these sentences.

In some sections of the manuscript, there is no connection with the figures and tables. Thus, there are explanations which are produced by the word processor automatically, such as “As shown in Figure Error! Reference source not found”. Please check the manuscript top to down and make any necessary corrections in the manuscript.

Please provide more information, considering what is new in the proposed method and how your model is different from other models.

In some figures, it is very difficult to distinguish the labels. If possible, please provide figures in higher resolution. Additionally, instead of keeping 4 figures in one, decrease the number of figures that consist of only one or two figures in one.

The number of tables is too many tables. It is better to remove or merge some of them. If you underline or highlight in bold which values in some of the tables are the best, it will be easier to follow the obtained results.

Please provide limitations of the study considering the datasets used. Additionally, please provide some explanation about the usage of plant images from different places. Please provide some information on where the model fails.

It is not very clear how fast the model works on poor devices. If possible, please test and provide some information on how the model runs on small devices, considering the time.

Some terms are used differently. Please check the manuscript for similar terms and prefer only one term for consistency.

The number of references used is low for such a comprehensive and detailed study. If possible, a few more recent publications should be added.

Figures 5 to 9 appear to have most likely been taken from other references. If so, the sources should be cited below the figures. Additionally, the typographical errors in the figures should be corrected. For example, “Artentien” in Figure 8.

Reviewer 2 Report

Comments and Suggestions for Authors

The aim of this study is to develop a lightweight and efficient segmentation framework—Sparse-MoE-SAM—that integrates sparse attention and a mixture-of-experts decoder to enable accurate plant disease segmentation in resource-constrained environments. Despite the extensive contributions, the article needs some corrections.

1.A comparative analysis should be conducted with more recent studies in the literature review.

2.Please correct all figure and table references in the article; there are many “Error! Reference source not found.” errors in the current text.

3.The hyperparameter settings of the proposed model should be added in detail.

  1. Problems and solutions encountered during the training of MoE architecture should be expressed more clearly.
  2. If there are disease types for which the model performs poorly, analysis should be performed on these.
  3. The performance of the model on complex leaf occlusions should be detailed.
  4. The reason why the Gumbel-Topk selection strategy used in the attention mechanism is preferred should be more clearly justified.
  5. Experimental analyses that specifically evaluate the model's performance in low-light conditions should be increased.
  6. Example scenarios should be given regarding practical applications of the model on mobile devices.
  7. The novelty of the proposed methodology should be emphasized.
  8. The conclusions section should be improved and more detailed.
  9. The quality of figures (Figure 5,6,7, 8, and 9) of the paper needs to be improved.
  10. Explain more clearly the specific innovation of the two-stage MoE decoder compared to existing multi-stage or hierarchical segmentation approaches.

Reviewer 3 Report

Comments and Suggestions for Authors

overall

  1. This manuscript proposes an innovative architecture Sparse-MoE-SAM in the field of plant disease segmentation, integrating sparse attention, dual-stage MoE, and enhanced ASPP modules. The experimental design is rigorous and the data presentation is comprehensive.

Abstract

  1. The abstract has clear logic and complete structure, successfully presenting the problem background, innovative methods, experimental results and conclusions.
  2. The data of "mean Intersection-over-Union (mIoU) of 94.2%" and "23.7% computation reduction" should further indicate whether they are compared with the original SAM model or other baselines.

Introduction

  1. When introducing Sparse Attention and MoE technology, more previous research in the field of plant pathology can be added (for example, whether similar concepts have been applied to disease detection instead of NLP/COCO segmentation).
  2. The language of some paragraphs is slightly colloquial (such as "cuts out repetitive computation parts"), and it is recommended to unify it into formal academic terms.

Materials and Methods

  1. Tables and figures are marked as "Table Error! Reference source not found". The reference code needs to be corrected to avoid affecting the reviewer's perception.
  2. The introduction of Gumbel-Top-k Sparse Attention should briefly explain the design intuition behind it, such as whether there is spatial or semantic correlation evidence to support it.
  3. It is recommended to organize the Loss Function formula into a table in advance (such as a table or summary list) to facilitate readers to get a full picture..

Results & Discussion

  1. All table and figure numbers should be added (currently all are "Table Error! Reference...") for clear presentation.
  2. In the ablation study section, it is recommended to add whether there is statistical significance (e.g., p-value), especially when the IoU improvement is usually 0.7–1.4%.
  3. In the expert routing frequency analysis, the MoE gating weight distribution can be further visualized.
  4. More details can be added for application cases actually deployed on edge devices (such as which devices they actually run on).
  5. It is recommended to strengthen the comparative discussion with existing research, especially whether there are sparse attention architectures other than the SAM series that can compete with it.

Reviewer 4 Report

Comments and Suggestions for Authors

The manuscript presents Sparse-MoE-SAM, an innovative deep learning framework designed to achieve accurate and efficient plant disease segmentation, particularly in resource-constrained settings such as edge devices, drones, and smartphones. The paper addresses challenging problems in plant pathology computer vision, such as the high computational complexity of dense-attention models, natural sparsity in disease manifestation, and adverse field image conditions. The integration of sparse attention mechanisms, a two-stage mixture of experts (MoE) decoder, and a sparse-enhanced ASPP module is well motivated and convincingly evaluated across multiple datasets. Some comments are given as follows.

  1. Several referenced figures and tables are labeled "Error! Reference source not found," which hampers granular interpretation and review. Ensure all tables/figures are correctly referenced, included, and visually clear.

  2. The data availability statement is good, but code and pretrained model availability would significantly enhance reproducibility and community impact. Release code/scripts and, when possible, setup instructions on public repositories.

  3. While the ablation results attest to each module's contribution, consider more extensive analysis (e.g., varying the top-k in sparse attention and MoE independently, different gating network depths, or trade-offs with more/fewer expert branches).

  4. Visualizations of attention maps and expert routing with corresponding real-image samples would greatly help readers grasp spatial specializations and biological alignment.

  5. While three limitations are concisely mentioned, elaborating on concrete technical steps planned for hardware-level sparse optimization, meta-learning expert discovery, or integration of multimodal imaging could inspire follow-up research.

  6. For publication readiness, the manuscript would benefit from tighter editing to fix section references, streamline equations (numbering and formatting), and clarify domain-specific terminology for broader readership.

  7. A graphical summary of the architecture (without reference errors) would aid comprehension, especially for interdisciplinary audiences.

Round 2

Reviewer 1 Report

Comments and Suggestions for Authors

The authors have implemented almost all of the changes I suggested. However, the revision I requested regarding the figures in my comment number 4 has only been partially addressed, and the authors’ response was: “We acknowledge this concern and plan to provide higher resolution figures in the final version.” I do not find this response sufficient that the authors refer to the publication stage of a study which is not yet finalized and still under review as a justification for not making the requested correction. Therefore, I believe that the necessary revisions and improvements should be made as indicated in the relevant comment, and I evaluate this as aminor correction. 

Author Response

Comments 1:The authors have implemented almost all of the changes I suggested. However, the revision I requested regarding the figures in my comment number 4 has only been partially addressed, and the authors’ response was: “We acknowledge this concern and plan to provide higher resolution figures in the final version.” I do not find this response sufficient that the authors refer to the publication stage of a study which is not yet finalized and still under review as a justification for not making the requested correction. Therefore, I believe that the necessary revisions and improvements should be made as indicated in the relevant comment, and I evaluate this as aminor correction. 

Response 1:

We sincerely apologize for the lack of clarity in our previous response. In fact, during the major revision stage we had already improved the resolution of all figures in the manuscript. To ensure better clarity, we adopted two measures: (1) using LaTeX for manuscript preparation, which allowed us to insert the original images directly without compression, and (2) exporting all figures at higher resolution (1920×1080).

In addition to this overall enhancement:

Figures 1 and 2: we not only enhanced the resolution but also replaced several labels with clearer and more recognizable ones.

Figures 7 and 8: we enhanced the resolution and corrected typographical errors (e.g., “Artentien” corrected to “Attention”).

We thank the reviewer for pointing out this issue again and regret that our earlier response did not clearly convey the extent of the revisions already made.

Reviewer 2 Report

Comments and Suggestions for Authors

The authors have made all the necessary changes in the revised version of the manuscript.

Author Response

Comments 1:The authors have made all the necessary changes in the revised version of the manuscript.

Response 1:We sincerely thank the reviewer for the positive evaluation and for acknowledging the revisions. We greatly appreciate the constructive feedback provided during the review process, which has helped us improve the clarity, quality, and overall contribution of the manuscript.